# Know What You Don't Know:
# Uncertainty Calibration of Process Reward Models

**Young-Jin Park[1], Kristjan Greenewald[2], Kaveh Alim[1], Hao Wang[2,3], Navid Azizan[1]**

[1]Massachusetts Institute of Technology
[2]MIT-IBM Watson AI Lab
[3]Red Hat AI Innovation

## Abstract

Process reward models (PRMs) play a central role in guiding inference-time scaling algorithms for large language models (LLMs). However, we observe that even state-of-the-art PRMs can be poorly calibrated. Specifically, they tend to overestimate the success probability that a partial reasoning step will lead to a correct final answer, particularly when smaller LLMs are used to complete the reasoning trajectory. To address this, we present a calibration approach—performed via quantile regression—that adjusts PRM outputs to better align with true success probabilities. Leveraging these calibrated success estimates and their associated confidence bounds, we introduce an *instance-adaptive scaling* (IAS) framework that dynamically adjusts the compute budget based on the estimated likelihood that a partial reasoning trajectory will yield a correct final answer. Unlike conventional methods that allocate a fixed number of reasoning trajectories per query, this approach adapts to each instance and reasoning step when using our calibrated PRMs. Experiments on mathematical reasoning benchmarks show that (i) our PRM calibration method achieves small calibration error, outperforming the baseline methods, (ii) calibration is crucial for enabling effective IAS, and (iii) the proposed IAS strategy reduces inference costs while maintaining final answer accuracy, utilizing less compute on more confident problems as desired.

Project Page: http://young-j-park.github.io/know-what-you-dont-know

## 1   Introduction

Inference-time scaling is an emerging paradigm that improves the output quality of LLMs by trading off inference speed. Just as humans approach complex problems by breaking them down, reasoning through each step, and revising their thoughts, LLMs can also benefit from being given more time to "think" during inference [1]. While a myriad of inference-time scaling approaches have emerged, they often involve prompting the model with explicit reasoning instructions to encourage a structured thought process [51, 24]. Recent studies extend this idea by generating multiple candidate reasoning paths or answers and aggregating them to improve robustness [43]. Such strategies allow small-sized LLMs to achieve comparable (or even better) performance with large models when tackling challenging tasks, such as advanced mathematics or complex coding [20, 31].

An important component in many inference-time scaling algorithms is the chosen process reward model (PRM). PRMs are trained to quantify, as a reward, how good or desirable a model's intermediate-step outputs are with respect to a given task and/or to alignment with human preferences [6, 48, 30]. When normalized, reward scores are often interpreted as the probability that continuing from the current output will lead to a desirable outcome (e.g., producing a correct final answer in a math problem) [28]. In our experiments, we find, however, that even state-of-the-art PRMs can be

39th Conference on Neural Information Processing Systems (NeurIPS 2025).

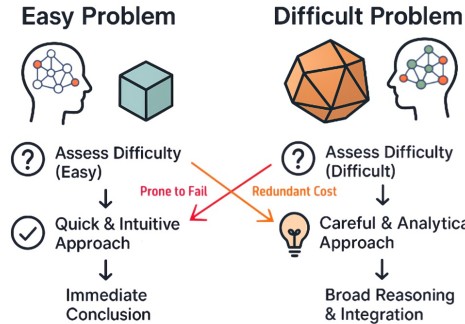

Figure 1: Why should problem-solving strategies adapt to task difficulty? Simple problems can often be solved through quick, intuitive solutions, whereas harder ones require extended, deliberate reasoning steps. Accordingly, it is desirable for LLMs to adjust their compute usage adaptively. Fixed-budget methods like best-of-$N$ are inefficient in this context: they either waste computation on easy tasks or fail to allocate sufficient resources to solve harder ones accurately.

miscalibrated, assigning overly optimistic scores—particularly on challenging, out-of-distribution problems. This is perhaps unsurprising since the probability of success depends on factors such as choice of model and inference method, which are not input into the PRM.

Miscalibration and overconfidence of PRMs can limit their utility beyond simply ranking partial reasoning steps. We highlight three important applications where calibrated PRMs are desirable: (1) providing interpretable uncertainty estimates to monitor LLM outputs during generation [58], (2) identifying when to say "I don't know" [see, e.g., 46, 7] or backtrack based on a low predicted probability of success, and (3) assessing the current problem difficulty or likelihood of success to adaptively adjust the inference-time compute budget. These applications motivate a fundamental question: **Can we calibrate off-the-shelf PRMs and leverage them to enhance inference-time scaling methods?**

In this paper, we introduce a pipeline for improving the calibration of any off-the-shelf PRMs, allowing their scores to more accurately reflect the *uncertainty* that a given LLM will reach the correct answer. We begin by showing that standard calibration techniques, such as temperature scaling [16], are inadequate for calibrating PRMs. To address this, we introduce a quantile regression [22] based scheme that reduces the calibration error of PRM scores. The resulting model predicts success probabilities, together with confidence bounds, for every query and intermediate reasoning step. Our method collects intermediate reasoning steps generated by a given LLM and completes them by independently rolling out full responses. This Monte Carlo rollout provides an empirical estimate of the true success probability for each prefix trajectory. Given this data, we fine-tune the PRM after replacing its prediction head with a quantile regression model. The resulting model produces reliable success probability estimates when evaluating intermediate steps in multi-step reasoning tasks.

We use these now-calibrated PRMs to enable *instance-adaptive scaling (IAS)*, a framework where test-time compute is adaptively scaled based on the calibrated PRM's assessment of difficulty/likelihood of success. Beyond saving compute, this concept has the potential to reduce overall latency for the user waiting for a response. The intuition mirrors how humans solve problems: we spend more time on difficult questions and allocate more effort to promising solution paths (see Figure 1). Similarly, IAS guides LLMs to invest more computation in challenging or high-potential reasoning paths. We focus on two widely used inference-time scaling methods: best-of-$N$ (BoN) and beam search (BS). We prove that, for any given query and reasoning trajectory, the calibrated reward score can estimate the minimum number of additional trajectories required to produce at least one correct answer. In short, our key contributions are:

- We address the overestimation of success probabilities in existing PRMs by introducing a calibration approach using quantile regression.
- We propose an instance-adaptive inference-time scaling strategy that dynamically allocates compute budgets by leveraging calibrated PRMs.
- We empirically validate our methods by: 1) demonstrating low calibration error, and 2) enhancing best-of-$N$ and beam search through instance-adaptive scaling, leading to improved efficiency.

## 1.1 Related Work

**Inference-time scaling.** LLM performance can be improved through multi-stage reasoning, which breaks problems into simpler subtasks [51, 24, 1]. Although more computationally expensive, techniques such as sampling multiple outputs and aggregating them substantially improve performance

and robustness [57, 43]. Methods such as majority voting, verifier-based selection [6, 30, 4], and reward-based Monte Carlo search [48, 50, 15] show strong gains in complex reasoning tasks. Yet, most focus on performance gains, with limited attention to its reliability and cost-effective utilization of computational resources.

**Process reward models.** Process reward models are specialized tools used in inference-time scaling that provide a step-by-step verification of LLMs' reasoning process, rather than evaluating only the final outcome [62]. Typically, PRMs are trained on datasets labeled at each reasoning step, either by humans or automated approaches like Monte Carlo rollouts and LLM-based judges. For example, Qwen-PRM [55] utilizes consensus-filtered labels from combined human and automated sources, achieving state-of-the-art accuracy [44], while Shepherd-PRM [50] effectively relies on purely automated labeling despite lower precision in detecting errors.

**Adaptive sampling for efficient state estimation.** Puri et al. [37] propose viewing LLM reasoning as a probabilistic state estimation problem. Note that the classical state estimation literature has explored methods for dynamically adjusting sample sizes based on state uncertainty [12, 13, 45, 10]. Similarly, information-driven adaptive strategies have been studied in the context of planning algorithms [18, 5, 29]. However, to the best of our knowledge, prior work has not proposed instance-adaptive sampling strategies within the context of LLM inference-time scaling.

For an extended discussion of related work, see Appendix A.

## 2 Preliminaries and Notation: Inference-Time Scaling with PRMs

We recall how reward models are used by standard inference-time scaling methods. Let LLM denote a language model that generates responses, and let PRM denote a process reward model used to evaluate the quality of the outputs produced by LLM. Given a query $q$, LLM generates a multi-step reasoning trajectory $\mathbf{x} = (x_1, x_2, \ldots, x_T)$, where $x_i$ represents the $i$-th reasoning step and $T$ is the total length of the trajectory. We denote the prefix of the trajectory up to step $t$ as $\mathbf{x}_{0:t}$ (in particular, $x_0 :=$" " and $x_{0:0}$ is an empty reasoning sequence). Below, we recap two standard PRM-based inference-time scaling methods.

**Best-of-$N$ (BoN)** [4]: Given a query $q$, we first use LLM to generate $N$ complete trajectories

$$\mathbf{x}^{(i)} = (x_1^{(i)}, x_2^{(i)}, \ldots, x_{T^{(i)}}^{(i)}) \sim \mathsf{LLM}(q), \quad \text{for } i = 1, \ldots, N.$$

Then we apply PRM to assign a score to each trajectory: $r^{(i)} = \mathsf{PRM}\left(q, \mathbf{x}^{(i)}\right)$. The final output is the trajectory with the highest reward $\mathbf{x}^{(i^*)}$ where $i^* = \arg\max_i r^{(i)}$.

**Beam Search (BS)** [43]: BS iteratively builds reasoning trajectories by alternating between generation and evaluation, rather than generating full sequences in a single pass. At step $t$, each of the $K$ surviving partial trajectories $\mathbf{x}_{0:t-1}^{(i)}$ is extended by one reasoning step. Specifically, for each trajectory, LLM generates $M$ candidate next steps: $\mathbf{x}_t^{(i,j)} \sim \mathsf{LLM}\left(q, \mathbf{x}_{0:t-1}^{(i)}\right)$ where $j = 1, \ldots, M$ denotes different continuations for the $i$-th trajectory. Then PRM evaluates each of the $N = K \times M$ extended trajectories: $r_t^{(i,j)} = \mathsf{PRM}\left(q, \mathbf{x}_{0:t}^{(i,j)}\right)$. Finally, among all $N$ candidates, only the top $K$ partial trajectories with the highest scores are kept for the next step. This pruning step ensures the search remains computationally tractable while focusing on promising reasoning paths.

## 3 Calibration and Process Rewards

PRMs are typically trained to quantify the quality of (partial) model outputs, i.e., their correctness, how well they advance the reasoning towards the solution to the query, and/or how well they align with human preferences. We highlight an additional, underexplored perspective: when normalized, PRM scores can be interpreted as an estimate of the probability that continuing from the current output will ultimately lead to a desirable outcome, such as producing a correct final answer in a multi-step reasoning task [50]. We observe that state-of-the-art PRMs often overestimate the chance of eventual success when interpreted in this way. We define the success probability as follows.

**Definition 1.** Given a query $q$ and a partial trajectory $\mathbf{x}_{0:t}$ generated so far, the *success probability* $p$ is the probability that autoregressively continuing this trajectory with a stochastic language model

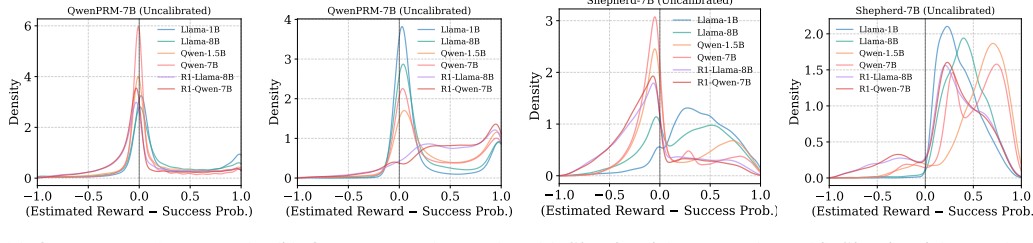

Figure 2: Histogram of signed deviations between PRM rewards (i.e., estimated success probabilities) and ground-truth success probabilities. Ground truth is estimated via Monte Carlo sampling: for each question and partial reasoning step prefix, we use a given LLM to generate multiple completions and compute the empirical success rate. We evaluate Qwen-PRM-7B and Shepherd-PRM-7B on the `MATH500` (in-distribution) and `AIME24-25` (out-of-distribution) datasets. Positive deviations indicate overestimation. PRMs consistently overestimate success probabilities, as evidenced by the distribution skewing right and/or peaking near $1.0$. This miscalibration is particularly pronounced for weaker completion models and more challenging, out-of-distribution problems.

(i.e., non-zero temperature), LLM, will ultimately yield a correct final answer.

$$p \triangleq \Pr\big(x_{t+1:T} \text{ generated by LLM yields a correct answer} \mid q, \mathbf{x}_{0:t}\big)$$

Here, we denote the index of a reasoning step by $t$ and the partial reasoning trajectory $\mathbf{x}_{0:t}$ is the sequence of outputs generated by the model from step $1$ to step $t \leq T$.

For the base case at $t = 0$, the partial trajectory $\mathbf{x}_{0:0}$ is an empty sequence, representing the state before any reasoning steps have been generated, where only the query $q$ is given. In this case, $p$ essentially estimates the difficulty of the query for LLM.

Since we have partial outputs, there is inherent uncertainty of future success (i.e., $p \in (0, 1)$), unlike single-turn settings where an output can definitively be evaluated for correctness. Note that this is a significant difference between our work and typical prior work on calibrated certainty estimates for LLMs, and is crucial to our quantile-based approach being feasible. That said, calibration metrics like Brier score and expected calibration error (ECE) still apply.

**The inherent calibration challenge of off-the-shelf PRMs.** Off-the-shelf PRMs lack a guarantee of calibration because their training is inherently dependent on a specific policy model. They learn a reward function from reasoning paths (rollouts) generated by one particular model (e.g., Qwen-Math-Instruct), tying their accuracy to that model's unique generative process and capabilities.

The problem originates from the autoregressive nature of language models, where sequence generation is policy-dependent. The probability of future tokens, $\pi_\theta(x_{t+1:T}|x_{0:t})$, is conditioned on the model's parameters, $\theta$. Consequently, *a PRM is only calibrated to the statistical patterns of the policy it was trained with*. Using it with a different policy introduces a distributional mismatch, breaking any calibration guarantee.

For instance, a PRM trained on a highly capable 72B model will overestimate the success probability when paired with a weaker 1B model. The PRM is calibrated to the stronger model's performance and cannot account for the weaker model's higher tendency to make errors on complex logical steps. This mismatch results in inflated and unreliable confidence scores, as presented in Figure 2.

**Why calibrate PRMs?** While we will show that a well-calibrated PRM is desirable for several important tasks, calibration is not required for the canonical task of ranking for BoN or BS, since relative ordering is all that matters. Specifically, any monotonic transformation of the true success probability preserves the ranking but changes calibration. As noted in the introduction, however, calibrated probabilities are essential for interpretability, systematic safety monitoring, and efficient budget allocation. In particular, calibration enables *instance-adaptive scaling* (IAS; discussed further in Section 4), where we dynamically adjust the number of trajectories to keep based on the estimated likelihood of success. In this setting, accurate probability estimates are essential for balancing efficiency and performance: a well-calibrated PRM allows us to reduce computations and maintain output quality. Furthermore, using a PRM rather than an outcome reward model is critical here, as we

need to estimate success probabilities before complete solutions exist. For beam search, we evaluate each partial reasoning prefix to guide search decisions. For best-of-$N$, we evaluate only the question itself (with no prefix) to determine the optimal sampling budget.

## 3.1 Calibration Data Collection Methodology

Now, we present a calibration-via-finetuning approach to mitigate the aforementioned issues. To calibrate a PRM for a given LLM, we construct a validation dataset through a systematic multi-stage generation and evaluation process. Please see details in Appendix C.1.

Our validation set is defined as $\cup_{q \in \mathcal{Q}_{\text{val}}} \{\mathbf{x}^{(q,1)}, \mathbf{x}^{(q,2)}, \ldots, \mathbf{x}^{(q,N_{\text{val}})}\}$, where $\mathcal{Q}_{\text{val}}$ represents a curated set of validation questions; in our experiments, we sample 500 random questions from MATH training split. For notational simplicity, we omit the superscript $q$ in the subsequent discussion, with the understanding that all operations are performed independently for each question in $\mathcal{Q}_{\text{val}}$.

**Stage 1: Initial Trajectory Generation.** For each validation question $q \in \mathcal{Q}_{\text{val}}$, we generate $N_{\text{val}} = 8$ independent reasoning trajectories using the target LLM. This yields a set of complete reasoning chains: $\{\mathbf{x}^{(1)}, \mathbf{x}^{(2)}, \ldots, \mathbf{x}^{(N_{\text{val}})}\}$. These trajectories represent diverse solution paths the model naturally explores for the given question.

**Stage 2: Prefix Extraction and Monte Carlo Rollouts.** For each reasoning trajectory $\mathbf{x}^{(i)} = x_{0:T^{(i)}}^{(i)}$ generated in Stage 1, we consider all possible prefix trajectories $x_{0:t}^{(i)}$, where $t \in \{0, 1, \ldots, T^{(i)}\}$ ranges from the initial step to the final step of the trajectory. For each prefix $x_{0:t}^{(i)}$, we perform a Monte Carlo estimation by generating $N_{\text{MC}} = 8$ additional follow-up trajectories. These follow-up trajectories are conditioned on both the original question $q$ and the specific prefix $x_{0:t}^{(i)}$, allowing us to estimate the probability of eventual success from that intermediate reasoning state.

**Stage 3: Success Probability Estimation.** For each prefix trajectory $x_{0:t}^{(i)}$, we evaluate all $N_{\text{MC}}$ follow-up trajectories to determine correctness. Let $Z^{(i,t)}$ denote the count of correct completions among these follow-ups. We then compute the empirical success probability: $\tilde{p}^{(i,t)} = Z^{(i,t)}/N_{\text{MC}}$, which serves as our ground-truth label for calibrating the PRM's predicted rewards.

Our calibration datasets, consisting of $\left(q, x_{0:t}^{(q,i)}, \tilde{p}^{(q,i,t)}\right)$ triplets (question, prefix, and empirical success probability), are publicly available for various reasoning LLMs.[1]

## 3.2 Uncertainty-Aware Calibration through Quantile Regression

We seek to fine-tune PRMs to align their predictions with success rates $p^{(i,t)}$ given the input $q$ and generated steps so far $x_{0:t}^{(i)}$. However, predicting success probabilities is inherently uncertain: calibration data is non-exhaustive and model capacity is finite. As with any machine learning model trained via empirical risk minimization (ERM), PRMs make loss-minimizing predictions that approximate the *conditional mean* of the success probability.[2]

While the conditional mean is a reasonable point estimate, it presents a critical problem: for any given test case, there is roughly a 50% chance the true success probability falls below this estimate. This means standard PRM architectures will systematically *overestimate* success probability for half of all instances. For applications requiring conservative estimation—such as instance-adaptive scaling (IAS), where we allocate computational budget based on estimated difficulty—overestimation is particularly problematic. If we overestimate $p^{(i,t)}$, we allocate insufficient samples, leading to high failure rates on problems that genuinely require more computation.

Instead of predicting the conditional mean, we propose predicting a *lower quantile* of the posterior distribution over $p^{(i,t)}$. By targeting, say, the 10th percentile, we ensure that for at least 90% of test cases, our estimate is below or equal to the true success probability. This provides a conservative *lower bound* that protects against underallocation of compute while still enabling meaningful instance-adaptive decisions.

---

[1] https://huggingface.co/datasets/young-j-park/prm_calibration
[2] Exactly the conditional mean for MSE loss and approximately so for cross-entropy loss when the posterior variance is low.

To achieve this, we modify the PRM architecture to perform quantile regression. Specifically, we expand the output dimension of the prediction head to generate multiple predictions—one for each target quantile level $\beta_n$ (e.g., 10%, 50%, and 90%)—rather than a single reward value. Details of this architectural modification and initialization are in Appendix C.5.

We then fine-tune the model using a weighted quantile loss (wQL) [23]:

$$\text{wQL}(\hat{r}, \tilde{p}) \triangleq \frac{1}{N_q} \sum_{n=1}^{N_q} \left[ \beta_n \cdot \max\left( 0, \ \tilde{p} - \hat{r}^{(\beta_n)} \right) + (1 - \beta_n) \cdot \max\left( 0, \ \hat{r}^{(\beta_n)} - \tilde{p} \right) \right],$$

where $N_q$ is the number of quantiles, $\hat{r}^{(\beta_n)}$ is the modified PRM's prediction for the $\beta_n$-quantile, and $\tilde{p}$ is the empirical success rate. This asymmetric loss penalizes overestimation more heavily for lower quantiles and underestimation more heavily for higher quantiles, encouraging the model to learn the distinct quantile levels. Importantly, training requires only point estimates $\tilde{p}$ from Monte Carlo sampling—we do not need to estimate the full distribution of $p$.

## 4 Instance-Adaptive Inference-Time Scaling

Existing inference-time scaling methods typically allocate a fixed sampling budget $N$—for example, by generating $N$ complete reasoning trajectories in BoN. However, this approach can be inefficient: for challenging tasks, $N$ samples may be insufficient to produce a correct answer, while for easier ones, it may waste computation. Just as humans allocate more effort to harder problems and less to simpler ones, it is desirable for LLMs to adjust their compute usage adaptively. Note that in the present work, we (perhaps erring on the side of optimism) assign the maximum budget to the hardest problems, but in principle, we could also choose to say "I don't know" or route to a more capable model if the probability of success is too low (difficulty too high).

In this section, we explore how to allocate the sampling budget dynamically based on the likelihood that an intermediate reasoning path will ultimately yield the correct answer, which we use as a proxy for the difficulty of the query. We begin with the following proposition, which characterizes the theoretical minimum number of samples needed on a per-instance basis:

**Definition 2.** Fix a question–prefix pair and let $p \in [0, 1]$ be the success probability (see Definition 1), conditioned on the current generation, that a single continued trajectory sampled from the language model is correct. For a target probability $C \in (0, 1)$ we define the *independent-sampling sample-complexity*

$$N^\star(p, C) \triangleq \min\left\{ n \in \mathbb{N} : \ \Pr\left(\text{at least one out of } n \text{ trajectories is correct}\right) \geq C \right\}.$$

**Proposition 1.** *For every $p \in (0, 1)$ and $C \in (0, 1)$,*

$$\text{set} \quad N_{\text{IAS}}(p, C) \triangleq \frac{\log(1 - C)}{\log(1 - p)}, \quad \text{then} \quad \left\lceil N_{\text{IAS}}(p, C) \right\rceil \geq N^\star(p, C).$$

*In other words, if* PRM *could perfectly distinguish correct from incorrect reasoning, then selecting the best trajectory among $N_{\text{IAS}}(p, C)$ samples (i.e., "best-of-$N_{\text{IAS}}$") guarantees an average accuracy of (at least) $C$ for questions whose per-trajectory success probability is $p$.*

See Appendix B.1 for a proof. The key takeaway is that the required sample size $N_{\text{IAS}}$ scales inversely with $\log(1 - p)$. Intuitively, for a fixed target accuracy, "easier" questions (with larger $p$) require significantly fewer trajectories, offering compute savings. Thus, a well-calibrated PRM enables adaptive sampling by estimating $p$ and then using it to guide how many trajectories to generate.

As noted previously, *PRMs frequently overestimate LLMs' success probabilities—particularly for weaker models or on hard, out-of-distribution queries* (Figure 2). Relying on these inflated scores leads to suboptimal inference-time scaling by selecting an excessively small $N_{\text{IAS}}$, whereas calibration mitigates these pitfalls and enables principled, cost-effective decision making.

### 4.1 IAS: Calibrated Reward-based Instance-Adaptive Scaling

Now, we propose a framework that can dynamically scale inference-time computes using an instance-adaptive scaling (IAS) strategy with calibrated PRMs. Theoretical justifications are in Appendix B.2.

**BoN+IAS.** Rather than drawing a fixed sample size $N$ for best-of-$N$ decoding, our IAS framework adaptively determines the number of trajectories to generate based on the estimated difficulty of each problem. Specifically, we compute the minimum number of samples $N_{\mathrm{IAS}}$ needed to achieve the target correctness level $C$ given the PRM's estimated success probability $\hat{r}^{(\beta)}$:

$$N_{\mathrm{IAS}} \triangleq \min\{\lceil N_{\mathrm{IAS}}(\hat{r}^{(\beta)}, C)\rceil \ , \ N_{\max}\},$$

where $N_{\max}$ is the maximum budget constraint. For problems with high estimated success probability, $N_{\mathrm{IAS}}$ will be small, while challenging problems receive larger budgets up to $N_{\max}$.

**BS+IAS-of-$M$ (with fixed $K$).** In beam search with a fixed beam width $K$, we can adaptively determine how many continuations to sample per prefix at each step. Suppose at a given beam-search step we have $K$ candidate prefixes. Let $\hat{r}^{(\beta)}\mathrm{min}$ denote the minimum calibrated success probability among these $K$ prefixes. To ensure at least one correct completion across all $K$ prefixes with confidence level $C$, IAS computes the number of samples per prefix as:

$$M_{\mathrm{IAS}} \triangleq \min\left\{\left\lceil \frac{N_{\mathrm{IAS}}\left(\hat{r}^{(\beta)}_{\min}, C\right)}{K}\right\rceil \ , \ M_{\max}\right\}.$$

where $M_{\max}$ is the maximum number of expansions allowed per prefix. By using the most pessimistic estimate $\hat{r}^{(\beta)}_{\min}$, we ensure sufficient sampling even for the most challenging prefix in the beam.

**BS+IAS-of-$K$ (with fixed $M$).** Conversely, when the number of expansions per prefix $M$ is fixed, we can adaptively determine the beam width itself. After generating $M$ continuations for each of $K$ candidate prefixes and ranking them by their calibrated rewards $\hat{r}^{(\beta)}$, IAS determines how many top-ranked prefixes to retain:

$$K_{\mathrm{IAS}} \triangleq \min\left(\{K_{\max}\} \cup \{k \mid N_{\mathrm{IAS}}(\hat{r}^{(\beta)}_k, C) \leq k \times M, \quad 1 \leq k \leq K_{\max}\}\right)$$

where $\hat{r}^{(\beta)}_k$ is the calibrated reward of the $k$-th best prefix and $K_{\max}$ is the maximum allowed beam width. This ensures that the total budget $K_{\mathrm{IAS}} \times M \geq N_{\mathrm{IAS}}(\hat{r}^{(\beta)}_{K_{\mathrm{IAS}}}, C)$, maintaining the target confidence level $C$ that at least one prefix will lead to a correct answer.

**Selecting $\beta$ involves a trade-off:** prioritizing efficiency could suggest using the median or a higher quantile, whereas selecting a lower quantile (e.g., 10%) follows our motivation of ensuring that a specified probability of success is achieved. We can formalize this latter point as follows, using the framework of conformal prediction as applied to quantile regression in [38].

**Theorem 1.** *Set $N = \infty$. Let $\hat{r}^{(\beta)}$ be the prediction of the $\beta$th quantile, and suppose we have held out an $n$-sample validation set $\mathcal{V}_n$. Then, on test input $X_{n+1}$,*

$$P(\text{success best-of-}N_{\mathrm{IAS}}(\hat{r}^{(\beta)}, C)|X_{n+1}) \geq C(1 - \tilde{\beta})$$

*where $\tilde{\beta} \geq \beta$ depends on $\mathcal{V}_n$ (see Appendix B.2.1 for its specification).*

## 5 Numerical Experiments

We evaluate our method on two mathematical-reasoning benchmarks—`MATH500` [17] and `AIME24-25` (i.e., `AIME2024` and `AIME2025`)—using six LLMs: Llama-3.2-1B & 3.1-8B-Instruct [47], Qwen2.5-Math-1.5B & 7B-Instruct [55], DeepSeek-R1-Distill-Llama & Qwen-8B [9].

We use Qwen2.5-Math-PRM-7B [62] as the primary PRM throughout the main manuscript, unless otherwise specified, as it was the top-performing open-source small-sized PRM in PRMBench [44]. We present experimental details and additional results for ReasonEval-7B [52] and Math-Shepherd-Mistral-7B [50] PRMs, along with comprehensive analyses in Appendices D and E, respectively.

### 5.1 Fine-Tuning PRMs for Better Calibration

We evaluate calibration errors of off-the-shelf PRMs and then show how our fine-tuning strategy reduces these errors. To start with, we construct a calibration dataset by randomly sampling 500 questions from the `MATH` training set. We assess calibration performance using standard metrics: Brier score [3], positive Brier score (i.e., the mean square of overestimation error, $\max\{\hat{y} - y, 0\}$),

Table 1: Calibration error before and after applying our calibration-via-finetuning method. We evaluate on the `MATH500` (in-distribution) and `AIME24-25` (out-of-distribution) datasets, using various LLMs to generate responses. Four calibration error metrics are reported (lower is better); the worst and best values for each dataset are highlighted in red and blue, respectively. Results show that our method consistently improves PRM calibration across different models and datasets.

| Dataset | Model | Brier | | PosBrier | | AdaptiveCE | | ECE | | AverageCE | |
|---|---|---|---|---|---|---|---|---|---|---|---|
| | | Uncal. | Calib. | Uncal. | Calib. | Uncal. | Calib. | Uncal. | Calib. | Uncal. | Calib. |
| MATH500 | Llama-3.2-1B | 0.2414 | 0.0692 | 0.2226 | 0.0472 | 0.2830 | 0.0811 | 0.2791 | 0.0942 | 0.3130 | 0.1840 |
| | Llama-3.1-8B | 0.2045 | 0.1210 | 0.1771 | 0.0994 | 0.2625 | 0.1674 | 0.2368 | 0.1515 | 0.2048 | 0.1876 |
| | Qwen-2.5-1.5B | 0.1541 | 0.1271 | 0.1305 | 0.1072 | 0.2176 | 0.1545 | 0.1554 | 0.1414 | 0.1229 | 0.1462 |
| | Qwen-2.5-7B | 0.1008 | 0.0818 | 0.0846 | 0.0527 | 0.1459 | 0.0999 | 0.0981 | 0.0864 | 0.0920 | 0.1227 |
| | R1-Llama-8B | 0.1614 | 0.0888 | 0.1140 | 0.0546 | 0.2505 | 0.1128 | 0.1311 | 0.0905 | 0.1517 | 0.1094 |
| | R1-Qwen-7B | 0.1480 | 0.0828 | 0.1056 | 0.0578 | 0.2413 | 0.1066 | 0.1095 | 0.0857 | 0.1957 | 0.1130 |
| AIME24-25 | Llama-3.2-1B | 0.1936 | 0.0029 | 0.1918 | 0.0005 | 0.2364 | 0.0108 | 0.2364 | 0.0041 | 0.4921 | 0.1306 |
| | Llama-3.1-8B | 0.2274 | 0.0414 | 0.2227 | 0.0354 | 0.2839 | 0.0862 | 0.2839 | 0.0782 | 0.4580 | 0.3988 |
| | Qwen-2.5-1.5B | 0.3302 | 0.0727 | 0.3220 | 0.0528 | 0.4007 | 0.1054 | 0.4007 | 0.0865 | 0.3936 | 0.2889 |
| | Qwen-2.5-7B | 0.2894 | 0.0721 | 0.2820 | 0.0657 | 0.3547 | 0.0982 | 0.3547 | 0.0982 | 0.3892 | 0.2829 |
| | R1-Llama-8B | 0.3846 | 0.0782 | 0.3712 | 0.0296 | 0.5259 | 0.1275 | 0.4764 | 0.0761 | 0.3614 | 0.1566 |
| | R1-Qwen-7B | 0.4144 | 0.0694 | 0.4018 | 0.0338 | 0.5575 | 0.0898 | 0.5078 | 0.0689 | 0.3680 | 0.1261 |

Figure 3: Comparison of our calibration method with popular techniques—temperature scaling, isotonic regression, and histogram binning—on `MATH500` and `AIME24-25`. As shown, our quantile regression (QR) method reduces calibration error more effectively than these baselines.

AdaptiveCE [34], ECE [32], and AverageCE [33] between ground-truth Monte Carlo success rates and the PRMs' (median) predictions.

**Off-the-shelf PRMs are overconfident.** We examine the histogram of the deviations of the estimated reward from the true success probability ($\hat{r}^{(i,t)} - \tilde{p}^{(i,t)}$). As shown in Figure 2, the error densities are skewed to the right, with minimal mass on the left, indicating consistent overestimation. This effect is further amplified when weaker LLMs are used to generate responses, or when evaluated on more challenging datasets such as `AIME24-25`.

**Significance of calibration.** Table 1 presents numerical evidence supporting our earlier claim that PRMs frequently suffer from poor calibration and tend to overestimate. In contrast, our proposed calibration approach effectively mitigates these calibration issues. Additionally, the approach substantially improves results on out-of-distribution datasets, underscoring its effectiveness across diverse conditions. This finding further indicates that calibrated PRMs are better at differentiating more challenging questions or reasoning tasks from those with higher certainty.

**Calibration baselines.** We evaluate several standard calibration methods for calibrating PRMs, including temperature scaling [16], isotonic regression [61], and histogram binning [60]. Detailed descriptions of these methods are provided in the Appendix C.4. Figure 3 compares our calibration-via-finetuning method with these baseline techniques. Since baseline methods correct calibration by uniformly shifting or rescaling the output probabilities (or logits) of PRMs, they are effective only when every instance's prediction is consistently overestimated or underestimated. For instance, weaker models like Llama-3.2-1B often benefit from simple methods that reduce overall overconfidence. However, these techniques typically fail to calibrate well on out-of-distribution tasks, such as the `AIME24-25` dataset. In contrast, our method leverages contextual information—such as question categories and the position of the current reasoning step—to achieve robust calibration across diverse models and dataset distributions.

## 5.2 Calibrated Reward Enables Instance-Adaptive Scaling

We demonstrate IAS by applying it to two simple yet widely used inference scaling techniques: best-of-$N$ and $N$-beam search on steps.

Table 2: Comparison between the best-of-$N$ method using a fixed-$N$ strategy (BoN) and our proposed instance-adaptive sampling strategy (BoN+IAS). We report both accuracy and relative computational cost (budget), measured by the average number of samples per question normalized by $N = 64$: $\text{Budget} = N_{\text{IAS}}/N$. Relative improvements over Pass@1 accuracy are highlighted in light blue. Our IAS with calibrated PRMs achieves substantial compute savings without significant performance loss. Crucially, the effectiveness of IAS depends strongly on PRM calibration quality: uncalibrated PRMs tend to overestimate success probabilities, causing overly optimistic downsampling. Moreover, uncalibrated PRM's model-independent design prevents it from offering adaptive strategies for different LLMs, resulting in further degradation for weaker Llama models.

| Dataset | Model | Baselines | | w/ Uncal. PRM | | w/ Calib. PRM | |
| | | Pass@1 | BoN | BoN+IAS | Budget Ratio | BoN+IAS | Budget Ratio |
|---|---|---|---|---|---|---|---|
| MATH500 | Llama-3.2-1B | 0.2255 | 0.4760 | 0.2278 | 0.0162 | 0.4623 | 0.6381 |
| | Llama-3.1-8B | 0.4659 | 0.6440 | 0.4674 | 0.0162 | 0.6223 | 0.3631 |
| | Qwen-2.5-1.5B | 0.6970 | 0.7660 | 0.6973 | 0.0162 | 0.7537 | 0.2461 |
| | Qwen-2.5-7B | 0.7994 | 0.8540 | 0.7993 | 0.0162 | 0.8368 | 0.2342 |
| | R1-Llama-8B | 0.6734 | 0.8240 | 0.6729 | 0.0162 | 0.8042 | 0.3439 |
| | R1-Qwen-7B | 0.7556 | 0.8640 | 0.7569 | 0.0162 | 0.8568 | 0.3133 |
| AIME24-25 | Llama-3.2-1B | 0.0042 | 0.0000 | 0.0040 | 0.0195 | 0.0167 | 1.0000 |
| | Llama-3.1-8B | 0.0268 | 0.0500 | 0.0273 | 0.0195 | 0.0333 | 0.9685 |
| | Qwen-2.5-1.5B | 0.0932 | 0.1500 | 0.0962 | 0.0195 | 0.1522 | 0.9372 |
| | Qwen-2.5-7B | 0.0885 | 0.1167 | 0.0920 | 0.0195 | 0.1885 | 0.9099 |
| | R1-Llama-8B | 0.0784 | 0.1833 | 0.0838 | 0.0195 | 0.1217 | 0.9661 |
| | R1-Qwen-7B | 0.1411 | 0.2667 | 0.1425 | 0.0195 | 0.1795 | 0.9635 |

**Instance-adaptive BoN.** We evaluate the BoN framework ($N_{\max} = 64$) using both the uncalibrated and calibrated PRMs, comparing their performance against our proposed IAS strategy (BoN+IAS). We adopt a conservative setting with $C = 0.99$ and $\beta = 0.1$ (see Appendix E.5 for an ablation study). For a fair comparison, we use the calibrated PRM to determine $N_{\text{IAS}}$ and the original PRM for ranking after adaptive sampling. We report the accuracy of each method and the relative computational cost associated with IAS, the average number of samples per question normalized by $N_{\max}$.

Table 2 shows IAS, when combined with calibrated PRMs, *achieves significant computational savings while maintaining performance close to fixed-$N$ strategies*. More importantly, as previously discussed, the original PRMs often severely overestimate success probabilities, especially for smaller models (e.g., Llama-3.2-1B) or challenging tasks (e.g., AIME24-25). As a result, IAS reduces budgets too aggressively, leading to performance sacrifice. In contrast, our calibrated PRMs allow IAS to reduce compute more conservatively, preserving accuracy while saving computational cost.

Our findings also show that to effectively reduce computational cost, adaptive sampling requires a well-calibrated PRM. Uncalibrated PRMs, which often overestimate success, lead the BoN+IAS framework to reduce computational budgets too aggressively, resulting in degraded performance.

**Instance-adaptive Beam Search.** To further verify the efficacy of the IAS for intermediate reasoning steps in addition to the initial stage, we test IAS-of-$K$ (IASoK) and IAS-of-$M$ (IASoM), with a beam search setup ($N = 64$, $M = 8$, $K = 8$). We use the same setting of $C = 0.99$ and $\beta = 0.1$.

As shown in Table 3, our IAS strategy maintains accuracy while reducing the budget up to about 75%. The IASoM variant tends to be more conservative than ISAoK, yet often achieves higher accuracy with a comparable budget usage.

**Why do we need an instance-adaptive scaling?** Recall that our goal is not to introduce a new inference-time scaling method that universally outperforms existing approaches. Instead, we aim to enable inference-time scaling to allocate compute budgets dynamically based on a model's estimated likelihood of answering correctly. To this end, IAS adaptively determines, on a *per-instance* basis, the number of samples that best balance accuracy and computational cost.

Figure 4 reports the performance plot across varying question difficulty levels for different values of $N$, along with the resulting accuracy and cost of the proposed IAS strategy (see Appendix E.4 for full results). As expected, enlarging $N$ improves accuracy in a nearly monotonic fashion; nevertheless, the absence of a "sweet spot" renders the selection of a single budget inherently difficult. Furthermore, accuracy declines with increasing question difficulty, implying that the sample budget required to attain a target accuracy should be uncertainty-dependent. Consequently, choosing a fixed $N$ on a

Table 3: Comparison between the beam search (BS) method using a fixed-$N/M$, and our proposed adaptive sampling strategy (IASoK and IASoM) using calibrated PRMs. We report both accuracy and relative computational cost (budget) to the baseline, measured by the average number of LLM generations per question normalized by that of a fixed-budget BS baseline. Relative improvements over the Pass@1 accuracy are highlighted in light blue. The proposed IAS strategy yields substantial computational savings without significant performance loss.

| | | Baselines | | IAS w/ Calibrated PRM | | | |
| Dataset | Model | Pass@1 | BS | BS+IASoK | Budget Ratio | BS+IASoM | Budget Ratio |
|---|---|---|---|---|---|---|---|
| MATH500 | Llama-3.2-1B | 0.2255 | 0.5360 | 0.5180 | 0.8996 | 0.5320 | 0.9891 |
| | Llama-3.1-8B | 0.4659 | 0.6640 | 0.6500 | 0.5309 | 0.6740 | 0.6686 |
| | Qwen-2.5-1.5B | 0.6970 | 0.8060 | 0.7840 | 0.5528 | 0.8100 | 0.5993 |
| | Qwen-2.5-7B | 0.7994 | 0.8680 | 0.8540 | 0.5332 | 0.8560 | 0.5757 |
| | R1-Llama-8B | 0.6734 | 0.8140 | 0.8320 | 0.3581 | 0.8540 | 0.3962 |
| | R1-Qwen-7B | 0.7556 | 0.8280 | 0.8460 | 0.3345 | 0.8740 | 0.3652 |
| AIME24-25 | Llama-3.2-1B | 0.0042 | 0.0167 | 0.0167 | 1.0364 | 0.0167 | 1.0619 |
| | Llama-3.1-8B | 0.0268 | 0.0333 | 0.0667 | 0.2620 | 0.0500 | 0.5243 |
| | Qwen-2.5-1.5B | 0.0932 | 0.1333 | 0.1333 | 0.5027 | 0.1667 | 0.6097 |
| | Qwen-2.5-7B | 0.0885 | 0.2167 | 0.1500 | 0.6499 | 0.1833 | 0.7399 |
| | R1-Llama-8B | 0.0784 | 0.1333 | 0.2000 | 0.4210 | 0.3000 | 0.5209 |
| | R1-Qwen-7B | 0.1411 | 0.1333 | 0.2000 | 0.3640 | 0.3167 | 0.4464 |

convenient validation set may lead to pronounced performance deficits under distributional shift. For instance, tuning on Level 1 of the MATH500 benchmark would suggest a small $N$ regime, but this could be highly suboptimal for harder datasets.

IAS addresses this mismatch by adapting its budget to instance success probability: it expends roughly four times fewer samples on Level 1 items than on Level 5, while allocating additional samples where they are most needed. In doing so, IAS aligns computational expenditure with uncertainty, yielding superior cost-effectiveness without pre-defining a universally optimal $N$.

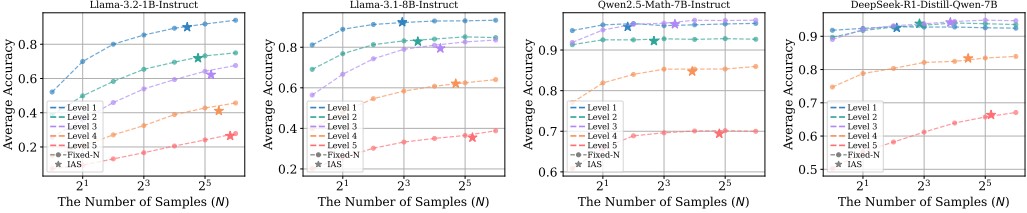

Figure 4: We illustrate average accuracy across test points of varying difficulty levels (1: easy to 5: hard). Results from the fixed-$N$ baseline and our instance-adaptive sampling (IAS) method are shown as dashed lines and stars, respectively. As shown, IAS dynamically adjusts sampling based on problem difficulty in MATH500, allocating more samples to harder tasks.

# 6   Conclusion and Limitations

This paper introduces a calibration strategy that enables PRMs to produce more reliable estimates of success probability: the likelihood that a partial reasoning step (or initial question) will lead to a correct final answer when completed by a given policy LLM. Through quantile regression, our approach provides conservative lower bounds rather than overoptimistic point estimates, addressing a key limitation of standard PRMs.

We demonstrate a practical application of well-calibrated PRMs in instance-adaptive inference-time scaling (IAS), where computational budget is allocated based on estimated success probability. By providing reliable estimates, our calibrated PRMs enable more efficient resource allocation, investing more computation on challenging problems while avoiding waste on easier ones.

Future work could explore fine-tuning strategies for better generalization across datasets and quantile levels, extend this approach beyond mathematical reasoning to domains like code generation and LLM agents, and investigate IAS in settings beyond best-of-N and beam search. We hope this work establishes a foundation for more reliable reward models and cost-effective adaptive scaling.

## Acknowledgments and Disclosure of Funding

This work was supported in part by the MIT-IBM Watson AI Lab, Jane Street, the MIT-Amazon Science Hub, the MIT-Google Program for Computing Innovation, and MathWorks.

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

# A    Extended Related Work

**Inference-time scaling.**    Recent studies have demonstrated that the capabilities of LLMs can be significantly enhanced by employing *multi-stage reasoning*, rather than directly generating answers in a single step [51, 24]. Although such methods typically require greater computational resources at inference time, performance improvements can be substantial by decomposing problems into simpler sub-tasks [1]. This strategy can be further improved by expanding the reasoning process: instead of relying solely on single-pass decoding, recent inference-time scaling techniques sample multiple candidate outputs and aggregate them for increased robustness [57, 43]. Those approaches—ranging from majority voting to verifier-based selection [6, 30, 4], as well as sophisticated Monte Carlo search algorithms [15] employing reward models [48, 50]—have demonstrated significant advances, particularly in reasoning-intensive tasks such as mathematical problem-solving. Nevertheless, most existing research has emphasized performance enhancement via extensive inference-time computation, while its reliability and cost-effective utilization of computational resources remain relatively understudied.

**Process Reward Models.**    PRMs are specialized inference-time scaling tools designed to enhance the reliability of LLMs by verifying each intermediate step of their reasoning processes, rather than evaluating only the final outcomes [62]. Training PRMs typically involves step-labeled datasets, which are generated either through detailed human annotation or via automated methods like Monte Carlo rollouts—sampling multiple solution trajectories to assess correctness probabilistically—and evaluations by other large language models acting as judges [55, 50]. Exemplifying this approach, Qwen-PRM [55] integrates a rigorous consensus-filtered labeling strategy that combines both human judgments and automated assessments, enabling it to achieve state-of-the-art accuracy in step-wise reasoning verification tasks [44]. Alternatively, Shepherd-PRM [50] illustrates that purely automated labeling approaches, while somewhat less precise in pinpointing individual reasoning errors, still significantly enhance overall model performance, showcasing their practicality and scalability. Recently introduced PRMs, such as ReasonEval [52], extend beyond basic validity checks by incorporating redundancy evaluation—assessing if reasoning steps are unnecessarily repetitive or redundant, thus further improving robustness and effectiveness in LLM inference.

**Adaptive sampling for efficient state estimation.**    Recent work by Puri et al. [37] suggests that LLM reasoning processes can be framed as particle filtering—*a sampling-based, probabilistic state estimation* approach. Classical state estimation literature has explored methods for dynamically adjusting sample sizes based on state uncertainty [12, 13, 45, 10]. The core principle behind

these adaptive sampling methods is to decrease sample sizes when uncertainty is low; this reduces computational costs while maintaining accuracy, particularly important for resource-constrained environments such as mobile robotics. Similarly, information-driven adaptive strategies have been studied in contexts of planning algorithms [18, 5, 29]. However, to the best of our knowledge, this paper represents the first exploration of instance-adaptive sampling strategies within the context of LLM inference-time scaling.

We would like to remark that Yu et al. [59] also tackles the adaptive reasoning problem, and their approach shares a similar spirit with ours. Their adaptation, however, is based on the initial success probability (i.e., question difficulty), whereas our method additionally adapts at intermediate reasoning steps (and thus is extendable to beam search). Specifically, our approach is enabled by calibrating PRMs, which allows us to estimate success probabilities throughout the multi-step reasoning process— a direction that has not been explored before. Furthermore, while they adapt the reasoning length, our method focuses on controlling the reasoning width (i.e., sample size), which we believe is a promising avenue to explore for integrating both approaches.

**Reliability of LLMs.** Although state-of-the-art inference-time scaling methods frequently rely on reward models, the reliability of these models has been relatively less explored. Johnson et al. [21] introduced the theoretical framework *Experts Don't Cheat*, proposing that reliable models generate predictions independent of paired hints when confident about their input; Yadkori et al. [54] further verified this idea extends to LLMs. Parallel research focuses on the concept of *semantic entropy* (SE) [25, 11], identifying hallucinations through contextual inconsistencies. Ye et al. [58] subsequently expanded SE to assess the reliability of process reward models. While these studies have made strides in quantifying LLM reliability, they do not specifically integrate reliability assessment into inference-time scaling strategies. Furthermore, rather than relying on PRM to estimate success probability (i.e., uncertainty), future work could explore uncertainty quantification tools (e.g., [14, 27, 40, 41, 36, 35]). There is also a line of calibration-oriented work on LLMs, such as Shen et al. [42], which presents another promising avenue to explore within our framework.

# B  Omitted Proofs

## B.1  Proof of Proposition 1

Here, we provide a proof of Proposition 1.

**Proposition 1.** *For every $p \in (0, 1)$ and $C \in (0, 1)$,*

$$\text{set} \quad N_{\text{IAS}}(p, C) \triangleq \frac{\log(1 - C)}{\log(1 - p)} \,, \quad \text{then} \quad \left\lceil N_{\text{IAS}}(p, C) \right\rceil \geq N^\star(p, C).$$

*In other words, if the PRM could perfectly distinguish correct from incorrect reasoning, then selecting the best trajectory among $N_{\text{IAS}}(p, C)$ samples (i.e., "best-of-$N_{\text{IAS}}$") guarantees an average accuracy of (at least) $C$ for questions whose per-trajectory success probability is $p$.*

*Proof.* Draw $n$ trajectories independently. The probability that none is correct is

$$\Pr[\text{no successes}] = (1 - p)^N.$$

Therefore, the probability of at least one success is

$$\Pr[\text{at least one success}] = 1 - (1 - p)^N.$$

We require

$$1 - (1 - p)^N \geq \delta \quad \Longleftrightarrow \quad (1 - p)^N \leq 1 - \delta.$$

Since $\log(1 - p) < 0$, taking logarithms we have

$$N \log(1 - p) \leq \log(1 - \delta) \quad \Longleftrightarrow \quad N \geq \frac{\log(1 - \delta)}{\log(1 - p)}.$$

Finally, setting $N = N_{\text{IAS}}(p, C) = \frac{\log(1-C)}{\log(1-p)}$, we have

$$N_{\min}(p, C) = \left\lceil \frac{\log(1 - C)}{\log(1 - p)} \right\rceil.$$

Finally, if one can sample $N$ independent trajectories each succeeding with probability $p$, and a perfect reward model always picks a correct one whenever it exists, then choosing

$$N = \left\lceil \frac{\log(1-C)}{\log(1-p)} \right\rceil$$

guarantees success probability at least $C$.

$\square$

We also note that a similar line of theoretical analysis has been explored in parallel by Schaeffer et al. [39].

## B.2 Theoretical Justifications of IAS with Beam Search

We now prove that the same "independent-sampling" bound $N_{\mathrm{IAS}}(p, C)$ underlies the *instance-adaptive scaling* strategies for beam search (BS+IAS). The key observation is that, if each sampled trajectory is independently correct with probability $p$ (conditional on the prefix), then total expansions under a given $K$ and $M$ provide $K \times M$ independent tries.

**Proposition 2** (BS+IAS-of-$M$). *Assume $K$ candidate prefixes, each of which independently yields a correct continuation with probability at least $p_{\min}$. Fix a target overall success probability $C \in (0, 1)$ and choose the per-prefix expansion width*

$$M = \left\lceil \frac{N_{\mathrm{IAS}}(p_{\min}, C)}{K} \right\rceil.$$

*Then expanding all $K$ prefixes by $M$ continuations (for a total of $K \times M$ trajectories) and selecting with a* perfect *reward model guarantees probability at least $C$ that* some *trajectory is correct.*

*Proof.* With $K \times M$ independent trials, the probability of zero successes is $(1 - p_{\min})^{KM}$. If $KM \geq N_{\mathrm{IAS}}(p_{\min}, C)$, then

$$(1 - p_{\min})^{KM} \leq (1 - p_{\min})^{N_{\mathrm{IAS}}(p_{\min}, C)} \leq 1 - C,$$

so the probability of at least one success is at least

$$1 - (1 - p_{\min})^{KM} \geq C.$$

$\square$

**Proposition 3** (BS+IAS-of-$K$). *After expanding each of $K_{\max}$ prefixes by $M$ continuations, let $p_1 \geq p_2 \geq \cdots \geq p_{K_{\max}}$. Define*

$$K_{\mathrm{IAS}} = \min\{k \mid N_{\mathrm{IAS}}(p_k, C) \leq k \times M, \ 1 \leq k \leq K_{\max}\}.$$

*Retaining the top $K_{\mathrm{IAS}}$ prefixes and discarding the rest ensures overall success probability $\geq C$.*

*Proof.* For the top $k$ prefixes, there are $k \times M$ independent trials, each succeeding with probability at least $p_k$. If $k \times M \geq N_{\mathrm{IAS}}(p_k, C)$, then

$$(1 - p_k)^{kM} \leq (1 - p_k)^{N_{\mathrm{IAS}}(p_k, C)} \leq 1 - C,$$

so the probability of at least one success is at least

$$1 - (1 - p_k)^{kM} \geq C.$$

$\square$

### B.2.1 Proof of Theorem 1

We require the following theorem from [38], which we slightly specialize to be one- instead of two-sided.

---

**Algorithm 1** One-Sided Split Conformal Quantile Regression (specialized from [38])

---

**Require:** Data $(X_i, Y_i) \in \mathbb{R}^p \times \mathbb{R}$, $1 \leq i \leq n$; Miscoverage level $\alpha \in (0,1)$; Quantile regression algorithm $\mathcal{A}$ (e.g. the QR method described in the main text[3]).

1: **procedure** SPLIT CONFORMAL QR
2:     Randomly split $\{1, \ldots, n\}$ into disjoint sets $\mathcal{I}_1$ and $\mathcal{I}_2$
3:     Fit quantile function: $\hat{q}^{(\beta)} \leftarrow \mathcal{A}(\{(X_i, Y_i) : i \in \mathcal{I}_1\})$
4:     **for** each $i \in \mathcal{I}_2$ **do**
5:         Compute $E_i = \hat{q}^{(\beta)}(X_i) - Y_i$.
6:     **end for**
7:     Compute $Q_{1-\alpha}(E, \mathcal{I}_2)$, the $(1-\alpha)(1 + 1/|\mathcal{I}_2|)$-th empirical quantile of $\{E_i : i \in \mathcal{I}_2\}$
8:     **return** Prediction interval $\mathcal{C}(x) = \left[\hat{q}^{(\beta)}(x) - Q_{1-\alpha}(E, \mathcal{I}_2), \infty\right)$ for $X_{n+1} = x$.
9: **end procedure**

---

**Theorem 2** (Theorem 1 of [38], specialized). *Suppose we have an exchangeable validation set $\mathcal{V}_n = \{(X_i, Y_i)\}_{i=1}^n$ where $X_i$ are features and $Y_i$ are labels. Consider a test point $X_{n+1}$ with unobserved true outcome $Y_{n+1}$. If $(X_i, Y_i)$, $i = 1, \ldots, n+1$ are exchangeable, then the prediction interval $C(X_{n+1})$ constructed by the (one-sided) Split Conformal Quantile Regression algorithm (Algorithm 1) satisfies*

$$P(Y_{n+1} \in \mathcal{C}(X_{n+1})) \geq 1 - \alpha(\mathcal{V}_n).$$

We can now prove the theorem.

*Proof of Theorem 1.* Follow the setting of Theorem 1, and consider that our exchangeable validation set $\mathcal{V}_n$ consists of data $(X_i, P_i)$ where $X_i$ are partial reasoning traces and $P_i$ are observed future probabilities of success.[4] We split $\mathcal{V}_n$ at random into $\mathcal{I}_1$ which we use for training $\hat{r}^{(\beta)}$, and $\mathcal{I}_2$.

Using Theorem 2 and Algorithm 1, we then have that for an (exchangeable) test point $(X_{n+1}, P_{n+1})$,

$$P(P_{n+1} \geq \hat{q}^{(\beta)}(X_{n+1}) - Q_{1-\alpha}(E, \mathcal{I}_2)) \geq 1 - \alpha. \tag{1}$$

Purely for notational convenience, let's set $\hat{q}^{(\beta)} = \hat{r}^{(\beta)} + \delta$ where $\delta$ and $\alpha := \tilde{\beta}$ are constants chosen using $\mathcal{I}_1$ such that $\delta \geq Q_{1-\tilde{\beta}}(E, \mathcal{I}_2)$ with high probability. This step is not strictly necessary; we simply use it to avoid having to adjust our existing quantile regressor. This guarantees that $\hat{q}^{(\beta)}(X_{n+1}) - Q_{1-\tilde{\beta}}(E, \mathcal{I}_2) \geq \hat{r}^{(\beta)}(X_{n+1})$, hence

$$P(P_{n+1} \geq \hat{r}^{(\beta)}(X_{n+1})) \geq 1 - \tilde{\beta}. \tag{2}$$

Now, consider that $P_{n+1}$ is the probability that 1 generation trial will succeed at recovering the correct answer, where here the probability is conditional on $X_{n+1}$, so independent of the above success probability. By definition, the probability that at least 1 of $N_{\text{IAS}}(P_{n+1}, C)$ trials will succeed is at least $C$, and $N_{\text{IAS}}(\cdot, C)$ is a monotonically decreasing function in its first argument. Therefore, by (2), with probability at least $1 - \tilde{\beta}$,

$$N_{\text{IAS}}(\hat{r}^{(\beta)}(X_{n+1}), C) \geq N_{\text{IAS}}(P_{n+1}, C),$$

and since probability of at least 1 out of $N$ success monotonically increases with $N$, with probability at least $1 - \tilde{\beta}$,

$$P(\text{success best-of-}N_{\text{IAS}}(\hat{r}^{(\beta)}(X_{n+1}), C)|X_{n+1}) \geq P(\text{success best-of-}N_{\text{IAS}}(P_{n+1}, C)|X_{n+1}) \geq C,$$

and therefore by independence,

$$P(\text{success best-of-}N_{\text{IAS}}(\hat{r}^{(\beta)}(X_{n+1}), C)|X_{n+1}) \geq C(1 - \tilde{\beta}).$$

$\square$

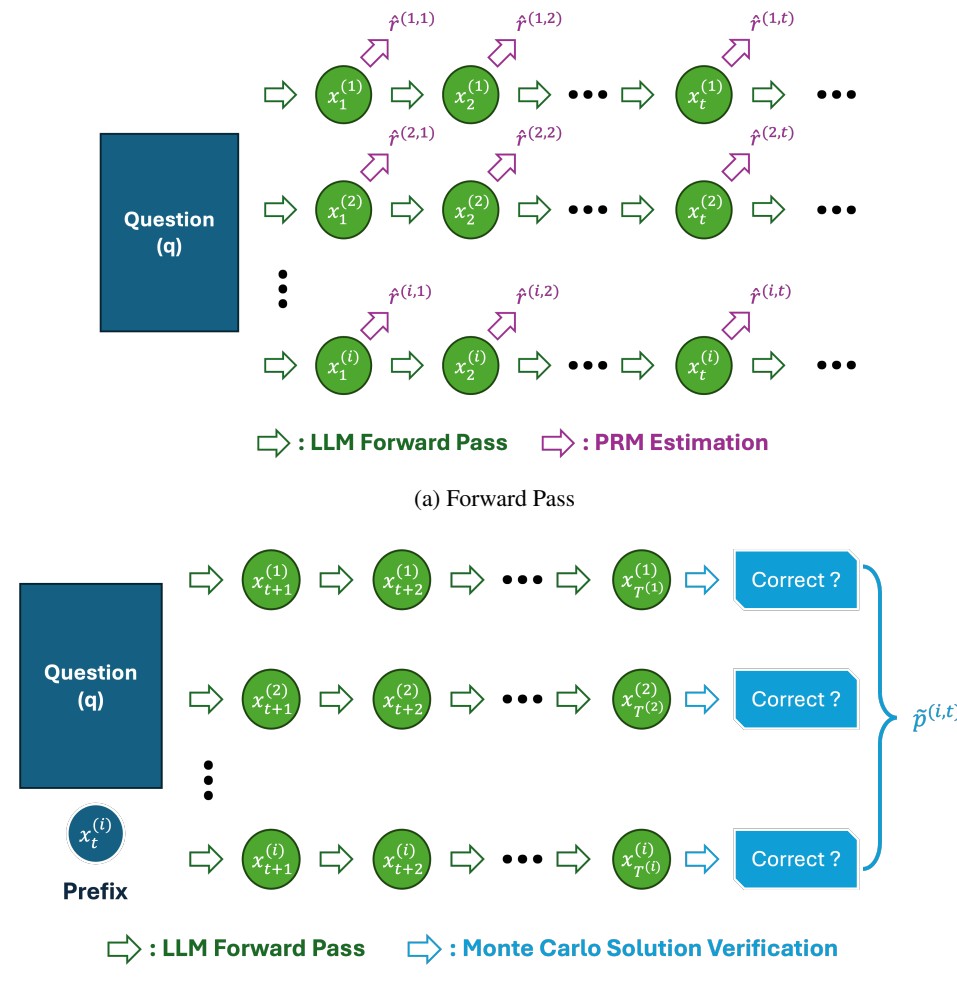

(a) Forward Pass

(b) Monte Carlo Estimation

Figure 5: For each validation $q$, we first generate independent reasoning trajectories for $i = 1, \ldots, N_{\text{val}}$. For each prefix trajectory, we conduct Monte Carlo simulations to estimate the success probability, $\tilde{p}^{(i,t)}$.

## C   Calibration via Fine-Tuning

### C.1   Calibration Set Collection

To evaluate the calibration quality of a given LLM and PRM pair, we construct a validation set $\mathcal{D}_{\text{val}} = \cup_{q \in \mathcal{Q}_{\text{val}}} \{(\hat{r}^{(q,i,t)}, \tilde{p}^{(q,i,t)})\}$ through a systematic three-stage process, as described in Section 3.1 of the main text. Here we provide additional implementation details and statistical analysis.

**Data Collection Procedure.**  For notational simplicity, we omit the superscript $q$ in the following discussion, with the understanding that all operations are performed independently for each question $q \in \mathcal{Q}_{\text{val}}$.

For each question $q \in \mathcal{Q}_{\text{val}}$, we first generate $N_{\text{val}} = 8$ independent reasoning trajectories using the target LLM: $\{\mathbf{x}^{(1)}, \mathbf{x}^{(2)}, \ldots, \mathbf{x}^{(N_{\text{val}})}\}$ (illustrated as green arrows in Figure 5a). Next, for each reasoning trajectory $\mathbf{x}^{(i)}$ and each of its prefix trajectories $x_{0:t}^{(i)}$ where $t \in \{0, 1, \ldots, T^{(i)}\}$, we generate $N_{\text{MC}} = 8$ additional follow-up trajectories conditioned on both the question and the prefix (green arrows in Figure 5b). By evaluating and counting the number of correct follow-up trajectories

---
[4]In practice, these may be noisy, but noise can be reduced with further Monte Carlo trials.

$Z^{(i,t)}$, we estimate the PRM's predicted reward $\hat{r}^{(i,t)} = \mathrm{PRM}_\phi(q, x_{0:t}^{(i)})$ and the empirical success probability $\tilde{p}^{(i,t)} \approx Z^{(i,t)}/N_{\mathrm{MC}}$ (purple arrows in Figure 5a and blue arrows in Figure 5b).

**Hardware and Software Configuration.** All simulations are conducted using NVIDIA V100 32GB SMX3 devices with the vLLM inference acceleration framework [26]. We employ the standard sampling configuration: `top_p = 1.0`, `top_k = -1` (considering all tokens in the vocabulary), and temperature $T = 0.7$ for all LLMs, consistent with best practices in recent reasoning model studies [2, 37].

## C.2 Statistical Properties and Label Quality

**Unbiased Estimation.** The per-prefix Monte Carlo estimator $\tilde{p}^{(i,t)}$ is statistically unbiased for the true success probability. While individual estimates with $N_{\mathrm{MC}} = 8$ have non-negligible variance ($\sigma_\varepsilon^2 = p(1-p)/N_{\mathrm{MC}} \approx 0.02$ for typical $p \approx 0.8$), our approach compensates through large-scale aggregation.

**Aggregate Accuracy.** We collect between $M = 40{,}000$ and $80{,}000$ prefix-label pairs per PRM–LLM combination. Aggregating over this large number of samples substantially reduces the standard error of our calibration metrics. Specifically, the standard error of the Brier score (our primary calibration measure) scales approximately as:

$$\mathrm{SE} \approx \sqrt{\frac{2\sigma_\varepsilon^4 + 4\sigma_\varepsilon^2 \cdot \mathrm{MSE}_{\mathrm{true}}}{M}}, \tag{3}$$

where $\sigma_\varepsilon^2 = \frac{p(1-p)}{N_{\mathrm{MC}}} \approx 0.02$ represents the per-instance variance (assuming typical success probability $p \approx 0.8$), and $\mathrm{MSE}_{\mathrm{true}} \approx 0.08$ represents the model's intrinsic prediction error. With our collection size of $M \geq 40{,}000$, this standard error becomes sufficiently small that our reported calibration metrics are statistically reliable, and the influence of per-instance noise on global trends is negligible.

**Impact on Quantile Regression.** While the aggregate statistics are accurate, the instance-wise standard error does artificially inflate the variance of the target distribution in our quantile regression objective. Increasing $N_{\mathrm{MC}}$ could potentially improve our method's performance by providing cleaner training signals. However, our experimental results demonstrate that our calibration approach is effective even with $N_{\mathrm{MC}} = 8$, achieving substantial improvements in calibration metrics across all evaluated PRM–LLM pairs.

## C.3 Additional Implementation Details

**Correctness Evaluation.** Following established validation procedures from prior works [56, 2], we determine correctness by parsing the generated answer within the `\boxed{}` delimiter and comparing it against the ground truth answer. Due to the mathematical nature of our evaluation datasets (MATH and related benchmarks), there is minimal ambiguity in correctness determination—answers are either numerically or symbolically equivalent to the ground truth or they are not.

**PRM Reward Calculation.** The reward computation follows the official implementation provided by each PRM developer. In our experiments, we extract the final score predicted at the last reasoning step, as this approach has been shown to perform well in inference-time scaling regimes [2, 43].

For Qwen-PRM, we use the following prompt format. The system prompt is:

```
<|im_start|>system
Please reason step by step, and put your final answer in \boxed{}.<|im_end|>
```

The user prompt format is:

```
<|im_start|>user
${Question}<|im_end|>
<|im_start|>assistant
${Prefix}<extra_0><|im_end|><|endoftext|>
```

where `$Question` represents the problem statement and `$Prefix` denotes the reasoning trajectory prefix (which is an empty string in the case when applying IAS for the Best-of-N regime). The reward score is computed based on the token probability at the special `<extra_0>` marker, which corresponds to the final reasoning step. Similar prompt structures are used for other PRMs, following their respective documentation.

**Data and Code Release.** To ensure full reproducibility and facilitate future research, we publicly release our complete codebase, including all prompt templates, sampling configurations, and evaluation scripts. Our calibration datasets—comprising $\left(q, x_{0:t}^{(q,i)}, \tilde{p}^{(q,i,t)}\right)$ triplets of question, prefix trajectory, and empirical success probability—are available for multiple reasoning LLMs at https://huggingface.co/datasets/young-j-park/prm_calibration. The dataset includes calibration data for all 18 PRM–LLM pairs evaluated in this work.

## C.4 Baselines

### C.4.1 Temperature Scaling

Typically, a PRM prediction head outputs two logits, e.g., for "*good*" vs. "*bad*" categories. Let these logits be denoted by $\ell_{\text{good}}$ and $\ell_{\text{bad}}$. Given a temperature parameter $T > 0$, the calibrated probability is computed as: $\hat{r} = \text{softmax}\left[\ell_{\text{good}}/T, \ \ell_{\text{bad}}/T\right]_1$, where the subscript 1 selects the "*good*" component. Choosing $T$ by minimizing a calibration metric (e.g., Brier score) on validation data can align $\hat{r}$ with the empirical success probability (to some extent).

### C.4.2 Isotonic Regression

Isotonic regression calibration fits a non-parametric, monotonic mapping from raw scores (e.g., PRM reward) to target values (e.g., success probabilities). The method applies the Pool Adjacent Violators algorithm (PAVA) [8] to learn a nondecreasing piecewise-constant function. This mapping then stretches or compresses score regions so that, within each fitted segment, the average predicted score matches the empirical success rate. Unlike parametric methods (e.g., temperature scaling), isotonic regression makes no assumptions about the shape of the calibration curve, allowing it to flexibly adapt to arbitrary monotonic distortions in the model's outputs.

### C.4.3 Histogram Binning

Histogram binning calibration partitions the prediction interval into a fixed number of equal-width bins, then replaces each raw score with the empirical success rate of its bin. At test time, each new prediction is assigned to its corresponding interval and is binned to that interval's mean. This simple non-parametric approach corrects systematic over- and under-confidence without assuming any particular shape of the calibration curve.

## C.5 Calibration via Fine-tuning

**Advantage of fine-tuning.** Since baseline methods correct calibration by uniformly shifting or rescaling the output probabilities (or logits) of PRMs, they are effective only when every instance's prediction is consistently overestimated or underestimated. In contrast, fine-tuning approaches leverage the capability of LLMs to capture contextual information such as question categories and the position of the current reasoning step, enabling more comprehensive calibration. For these reasons, we empirically observe that the simple baseline calibration methods are not adequate for PRM calibrations.

**Parameter-efficient fine-tuning.** To address the issue, we adopt a fine-tuning approach, arguably the most fundamental and straightforward method to calibrate the model. Specifically, we apply LoRA [19] with a rank of 2, a dropout rate of $0.1$, and a scaling factor of $32$. To further reduce the number of trainable parameters, we apply LoRA only for the query and value matrices in every fourth decoder layer as well as the prediction head.

The LoRA approach offers benefits for PRM calibration beyond its computational efficiency compared to full fine-tuning: it enables access to both the original and calibrated PRM scores in a single forward

pass. This allows for hybrid strategies (e.g., using the calibrated score for IAS and the original score for ranking candidate reasoning), without incurring significant additional computational cost.

**Quantile prediction.** For quantile regression, we modify the output head by replacing the final softmax linear layer with a sigmoid layer, producing outputs for each desired quantile (e.g., 3 quantiles corresponding to 10%, 50%, and 90% in our experiments).

To ensure the initial predictions remain consistent with the original softmax outputs, we initialize the sigmoid-based output head such that the sigmoid probability for each quantile matches the softmax probability of the "good" token from the original model.

More specifically, assuming a two-token prediction scenario with logits $z_0$ and $z_1$, then softmax probabilities for two classes 0 (i.e., "bad") and 1 (i.e., "good") are given by:

$$p(y = 1) = \frac{e^{z_1}}{e^{z_0} + e^{z_1}} \quad \text{and} \quad p(y = 0) = \frac{e^{z_0}}{e^{z_0} + e^{z_1}}$$

The probability for class 1 simplifies to a sigmoid function:

$$p(y = 1) = \frac{e^{z_1}}{e^{z_0} + e^{z_1}} = \frac{1}{1 + e^{-(z_1 - z_0)}} = \sigma(z_1 - z_0)$$

Thus, by equating the sigmoid input $z$ with the logit difference $z_1 - z_0$, we maintain consistency in initial predictions when transitioning from a softmax to a sigmoid-based output layer.

In practice, this corresponds to initializing the weights of the new linear sigmoid output layer $W$ as the difference between the original linear layer weights for the "good" token $W_1$ and the "bad" token $W_2$, i.e., $W \leftarrow W_1 - W_2$. Similarly, the bias term $b$, if it exists, is initialized as the difference of the biases from the original linear layers, i.e., $b \leftarrow b_1 - b_2$. This ensures the initial logit input to the sigmoid function directly matches the logit differences obtained from the original softmax-based model. Fine-tuning is performed using NVIDIA A100 40GB SXM4 GPUs and the TRL library [49].

## C.6 Model Calibration

Calibration error metrics assess whether a model's predicted probabilities are consistent with observed accuracy. Two widely used calibration metrics are the Brier score [3] and the expected calibration error (ECE) [32]. Brier score quantifies the mean-squared difference between predicted probabilities $\hat{y}^{(l)}$ and ground truth labels $y^{(l)}$ over $N$ instances, where $l = 1, \ldots, N$:

$$\text{Brier} \;=\; \frac{1}{N} \sum_{l=1}^{N} \big(\hat{y}^{(l)} - y^{(l)}\big)^2.$$

ECE evaluates the discrepancy between predicted confidence and empirical accuracy by partitioning predictions into $B$ confidence bins $\{\mathcal{B}_1, \ldots, \mathcal{B}_B\}$ and averaging the absolute difference within each bin. Denote the set of indices whose confidences fall into bin $\mathcal{B}_b$ by $\mathcal{I}_b$. Then

$$\text{ECE} \;=\; \sum_{b=1}^{B} \frac{|\mathcal{I}_b|}{N} \big|\text{conf}(\mathcal{B}_b) - \text{acc}(\mathcal{B}_b)\big|,$$

where $\text{conf}(\mathcal{B}_b) = \frac{1}{|\mathcal{I}_b|} \sum_{l \in \mathcal{I}_b} \hat{y}^{(l)}$ is the average confidence in bin $b$ and $\text{acc}(\mathcal{B}_b) = \frac{1}{|\mathcal{I}_b|} \sum_{l \in \mathcal{I}_b} y^{(l)}$ is the empirical accuracy. Variants of ECE include adaptive calibration error (AdaptiveCE) [34] and average calibration error (AverageCE) [33], which uses adaptive bin intervals and equal weighting across bins, respectivley.

# D Experiment Setup

We evaluate our method on two mathematical reasoning benchmarks: `MATH500` [17] and `AIME24-25` (i.e., `AIME2024` and `AIME2025`). `MATH500` is a 500-example subset of the MATH dataset, available at https://huggingface.co/datasets/HuggingFaceH4/MATH-500. For calibration purposes, we also constructed our own `MATH500-validation` set by randomly subsampling 500

problems from the original MATH training dataset, available at `https://huggingface.co/datasets/hendrycks/competition_math`. The AIME (American Invitational Mathematics Examination) consists of 15 questions per test, with two tests administered each year. Thus, we use 30 questions from 2024 and 30 from 2025, totaling 60 problems, available at `https://huggingface.co/datasets/HuggingFaceH4/aime_2024` and `https://huggingface.co/datasets/opencompass/AIME2025`, respectively.

To demonstrate the efficacy of our method across diverse LLMs, we evaluate six prominent models that vary in architecture, parameter count, and training paradigms (instruct-tuned vs. R1-distilled): Llama-3.2-1B and Llama-3.1-8B-Instruct [47], Qwen2.5-Math-1.5B and Qwen2.5-7B-Instruct [55], and DeepSeek-R1-Distill-Llama and DeepSeek-R1-Distill-Qwen-8B [9]. We use a standard temperature setting of 0.7 and adopt the inference prompt template specified in Puri et al. [37] across all models.

We use Qwen2.5-Math-PRM-7B [62] as the primary PRM throughout the main manuscript, as it was the top-performing open-source small-sized PRM in PRMBench [44]. We present additional results for ReasonEval-7B [52] and Math-Shepherd-Mistral-7B [50] PRMs, along with comprehensive analyses in the subsequent Appendix section. We adhered to the official prompting protocols for each PRM, as demonstrated on their respective websites.

We use NVIDIA V100 32GB SMX3 devices for every experiment except for quantile regression fine-tuning.

# E  Additional Results

## E.1  Calibration Analysis of Larger-Scale PRMs

We extend our calibration analysis to larger-scale process reward models by evaluating the 72B parameter variant on both `MATH500` and `AIME24-25` datasets. This analysis addresses whether the calibration challenges observed in 7B models persist at larger scales, and whether our calibration method remains effective across different model sizes.

Table 4 and Figure 6 present a comprehensive comparison of calibration performance across three configurations: uncalibrated 7B models, uncalibrated 72B models, and our calibrated 7B models. We evaluate six different base language models across two datasets, reporting both Brier score and Positive Brier score metrics for each configuration.

The results demonstrate that while larger 72B models exhibit reduced overestimation compared to their 7B counterparts, they still suffer from significant miscalibration issues. This observation is expected, as both model were trained with same policy model.

These results validate two key insights: (1) the calibration problem persists across model scales and cannot be solved simply by increasing model capacity, and (2) explicit calibration methods can effectively address miscalibration. Notably, our calibrated 7B model not only outperforms the uncalibrated 7B baseline but also surpasses the much larger uncalibrated 72B model, demonstrating that targeted calibration is more effective than naive scaling.

## E.2  Histogram of Estimation Error

We examine the histogram of the deviations of the estimated reward from the true success probability $(\hat{r}^{(i,t)} - \tilde{p}^{(i,t)})$, with both uncalibrated and calibrated PRMs.

As shown in Figures 7 and 8, the error densities of uncalibrated PRMs are skewed to the right, with minimal mass on the left, indicating consistent overestimation. This effect is further amplified when weaker LLMs are used to generate responses, or when evaluated on more challenging datasets such as `AIME24-25`.

It is noteworthy that native PRMs are designed to estimate success in a model-agnostic manner; consequently, they tend to overestimate success probabilities for weaker models, resulting in inflated calibration errors. However, calibration effectively mitigates the issue and transforms the PRM error histograms into approximately unbiased zero-mean distributions across different models and dataset distributions.

Table 4: Calibration performance comparison across model scales on MATH500 and AIME24-25 datasets (lower is better). Best results for each metric within each row are shown in **bold**. Results demonstrate that explicit calibration of Qwen-PRM-7B models outperforms even uncalibrated Qwen-PRM-72B models, highlighting the importance of targeted calibration over naive model scaling.

| Dataset | Model | Uncalibrated 7B | | Uncalibrated 72B | | Calibrated 7B | |
|---|---|---|---|---|---|---|---|
| | | Brier | Pos. Brier | Brier | Pos. Brier | Brier | Pos. Brier |
| MATH500 | Llama-3.2-1B | 0.241 | 0.223 | 0.211 | 0.078 | **0.069** | **0.047** |
| | Llama-3.1-8B | 0.205 | 0.177 | 0.177 | 0.111 | **0.121** | **0.099** |
| | Qwen-2.5-1.5B | 0.154 | 0.130 | 0.155 | **0.106** | 0.127 | 0.107 |
| | Qwen-2.5-7B | 0.101 | 0.085 | 0.118 | 0.057 | **0.082** | **0.053** |
| | R1-Llama-8B | 0.161 | 0.114 | 0.193 | 0.072 | **0.089** | **0.055** |
| | R1-Qwen-7B | 0.148 | 0.106 | 0.176 | 0.075 | **0.083** | **0.058** |
| AIME24-25 | Llama-3.2-1B | 0.194 | 0.192 | 0.157 | 0.063 | **0.003** | **0.001** |
| | Llama-3.1-8B | 0.227 | 0.223 | 0.183 | 0.122 | **0.041** | **0.035** |
| | Qwen-2.5-1.5B | 0.330 | 0.322 | 0.256 | 0.176 | **0.073** | **0.053** |
| | Qwen-2.5-7B | 0.289 | 0.282 | 0.205 | 0.139 | **0.072** | **0.066** |
| | R1-Llama-8B | 0.385 | 0.371 | 0.237 | 0.204 | **0.078** | **0.030** |
| | R1-Qwen-7B | 0.414 | 0.402 | 0.260 | 0.215 | **0.069** | **0.034** |

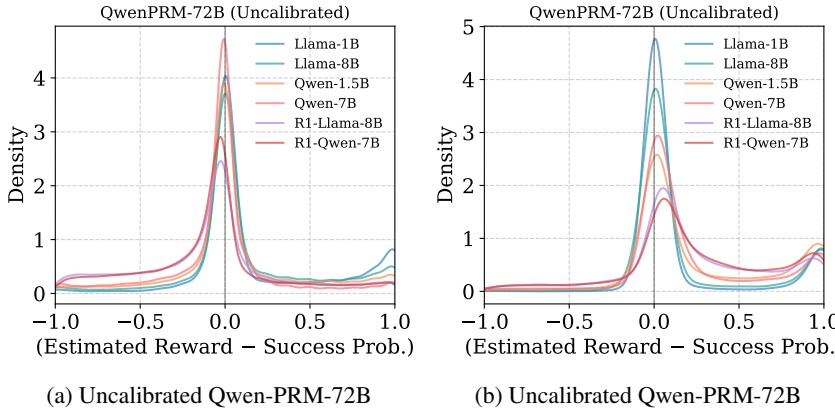

(a) Uncalibrated Qwen-PRM-72B  (b) Uncalibrated Qwen-PRM-72B

Figure 6: Histogram of signed deviation (i.e., estimation error) for Qwen-PRM-72B on the MATH500 (in-distribution) and AIME24-25 (out-of-distribution) datasets. Positive error indicates overestimation. While larger 72B model exhibit reduced overestimation compared to their 7B counterparts, they still suffer from significant miscalibration issues.

### E.3 Comparison with Calibration Baselines

In this section, we present calibration results for our method and several baselines across three PRMs. In addition to reporting the average Brier score, we also plot the 95% confidence intervals. The results are shown in Figures 9, 10, and 11.

The Shepherd PRM exhibits a trend similar to the Qwen PRM: our method consistently outperforms the baseline methods, whereas the baselines perform poorly on the out-of-distribution AIME24-25 dataset. The ReasonEval PRM, which is specifically trained to produce more robust outputs (especially for intermediate steps), appears to be already well-calibrated to some extent. Nevertheless, our method consistently surpasses the baseline approaches. Interestingly, despite its initially poor calibration, the Qwen PRM achieves better final calibration quality than both Shepherd and ReasonEval after fine-tuning.

We additionally compare our method against Adaptive Temperature Scaling (ATS) [53, 42], a recent calibration method that learns instance-specific temperature parameters. Since ATS was originally designed for standard language model outputs rather than PRM regression-style scores, we adapted it for our setting. Specifically, we modified ATS to train a transformer head that dynamically selects temperature parameters based on the PRM-generated hidden states of both the question and solution prefix, enabling context-aware calibration similar to our approach.

Table 5: Calibration error for various methods and models on MATH500 (in-distribution) and AIME24-25 (out-of-distribution). Values are presented as adaptive calibration error (AdaCE) and Brier scores, where lower is better. The best result for each metric within a row is **bolded**. We use Qwen-PRM-7B. Acronyms: TS (Temperature Scaling), ATS (Adpative Temperature Scaling), IR (Isotonic Regression), and HB (Histogram Binning).

| Dataset | Model | Uncalibrated | | TS | | ATS | | IR | | HB | | Calibrated (Ours) | |
|---|---|---|---|---|---|---|---|---|---|---|---|---|---|
| | | AdaCE | Brier | AdaCE | Brier | AdaCE | Brier | AdaCE | Brier | AdaCE | Brier | AdaCE | Brier |
| MATH500 | Llama-3.2-1B | .283 | .241 | .265 | .190 | .181 | .099 | .212 | .097 | .213 | .097 | **.081** | **.069** |
| | Llama-3.1-8B | .263 | .205 | .244 | .168 | .237 | .130 | .321 | .166 | .323 | .166 | **.167** | **.121** |
| | Qwen-2.5-1.5B | .218 | .154 | .205 | .140 | .205 | **.123** | .271 | .145 | .274 | .146 | **.155** | .127 |
| | Qwen-2.5-7B | .146 | .101 | .135 | .093 | .135 | **.076** | .201 | .092 | .205 | .094 | **.100** | .082 |
| | R1-Llama-8B | .251 | .161 | .223 | .149 | .196 | .107 | .297 | .141 | .298 | .141 | **.113** | **.089** |
| | R1-Qwen-7B | .241 | .148 | .202 | .140 | .178 | .103 | .289 | .132 | .289 | .132 | **.107** | **.083** |
| AIME24-25 | Llama-3.2-1B | .236 | .194 | .210 | .193 | .125 | .065 | .177 | .065 | .177 | .065 | **.011** | **.003** |
| | Llama-3.1-8B | .284 | .227 | .248 | .206 | .281 | .125 | .392 | .215 | .396 | .215 | **.086** | **.041** |
| | Qwen-2.5-1.5B | .401 | .330 | .393 | .293 | .367 | .184 | .477 | .331 | .486 | .337 | **.105** | **.073** |
| | Qwen-2.5-7B | .355 | .289 | .354 | .249 | .271 | .146 | .390 | .250 | .404 | .261 | **.098** | **.072** |
| | R1-Llama-8B | .526 | .385 | .528 | .310 | .438 | .220 | .590 | .400 | .588 | .399 | **.128** | **.078** |
| | R1-Qwen-7B | .558 | .414 | .570 | .340 | .440 | .230 | .613 | .430 | .613 | .430 | **.090** | **.069** |

Table 5 presents comprehensive comparisons across three calibration metrics. The modified ATS consistently outperforms simpler baselines, validating the importance of context-aware calibration. On the in-distribution MATH500 dataset, ATS achieves competitive performance—for instance, reducing AdaCE from 0.241 to 0.178 for R1-Qwen-7B. However, our method achieves superior calibration with AdaCE of 0.107 for the same model.

The performance gap becomes more pronounced on the out-of-distribution AIME24-25 dataset. While ATS provides improvements over uncalibrated models (e.g., reducing AdaCE from 0.558 to 0.440 for R1-Qwen-7B), our method demonstrates substantially better generalization with an AdaCE of only 0.090. This pattern holds consistently across all models and metrics. For example, on Llama-3.2-1B, our method achieves an AdaCE of 0.011 compared to 0.125 for ATS, and a Brier score of 0.003 compared to 0.065.

### E.4 Inference-time Scaling with IAS

We first reiterate that our goal is "not" to introduce a new inference-time scaling method that universally outperforms existing approaches. Instead, we aim to enable inference-time scaling to allocate compute budgets dynamically based on a model's estimated likelihood of answering correctly. The proposed IAS strategy enables us to adaptively determine, on a *per-instance* basis, the number of samples that best balance accuracy and computational cost.

**IAS for best-of-$N$.** We first present results and analysis of the proposed IAS strategy across different LLMs and PRMs. Specifically, we report performance under the best-of-$N$ framework in Tables 2, 6, and 7. To ensure statistical significance, the pass@1 rate is estimated by averaging accuracy over 64 independent forward passes. Similarly, for the IAS approach, we report the score based on 100 different trials for each question, based on the determined $N_{\text{IAS}}$ value computed individually per question.

As observed, the IAS approach adaptively allocates the sampling budget while largely preserving overall accuracy. Ironically, the ReasonEval PRM performs poorly under the best-of-$N$ (BoN) strategy, often yielding results worse than the pass@1 rate. However, this is not a limitation of the IAS strategy itself, but rather a reflection of the ReasonEval PRM's weakness in selecting the correct answer from among candidate responses. Notably, the budgets determined by IAS for ReasonEval are comparable to those derived from the other PRMs (See Figures 12, 13, and 14 for how effectively IAS adaptively selects $N$.). In contrast, PRMs like Qwen exhibit strong ranking capabilities despite poor initial calibration. Our calibration method proves particularly effective in such cases, as it preserves the PRM's original strengths while enabling effective IAS and improving its reliability.

**IAS for beam search.** To further demonstrate the effectiveness of the proposed IAS approach, we evaluate it under a beam search setup, which requires PRM evaluation over intermediate reasoning trajectories. However, due to limitations of the vLLM acceleration framework—which does not support fine-tuned custom models or provide access to internal states—inference was significantly slower than simple forward passes. In our setting ($N = 8 \times 8 = 64$, $M = 8$) and computation resource (i.e., V100 gpus), a single search over 500 examples took approximately 10 hours, resulting

in each experiment requiring over 200 hours to complete. To mitigate this computational burden, we adopted a simplified approach: instead of performing beam search at every step, we applied it every five steps. Given that typical reasoning trajectories span 20 or more steps, this amounted to roughly five beam search applications per example. Due to the computational burden, we were only able to perform a single trial for each beam search experiment.

Tables 3, 8, and 9 present the pass@1 accuracy for standard beam search (BS), BS with the proposed IASoK, and BS with IASoM, evaluated using the Qwen, Shepherd, and ReasonEval PRMs. For the Shepherd and ReasonEval PRMs, we conduct experiments using the `MATH100` (the first 100 examples among `MATH500`) and `AIME2024` datasets. To ensure a fair comparison, all experiments were conducted using the calibrated PRM. The results are consistent with the observations made for the Qwen PRM, as discussed in the main text.

### E.5    Ablation Studies on IAS Parameters

We provide an ablation study on two hyperparameters that govern the trade-off between computational cost and accuracy, under the best-of-$N$ setup.

**A target probability ($C$).**    The target probability serves as a parameter that acts as a constant multiplier on the expression $1/\log(1-p)$. We present results for values of $C$ such that $\log(1-C)$ takes on values $-0.125$, $-0.25$, $-0.5$, $-1.0$, $-2.0$, $-3.0$, $-4.0$, and $-5.0$. Figure 15 shows the budget-accuracy trade-off plots. We use $C = 0.99$, as it appears to provide a reasonable balance between accuracy and computational cost.

**A quantile parameter ($\beta$).**    The parameter $\beta$ represents a quantile that controls how conservatively we estimate the success probability. We report results for $\beta = 0.1$, 0.5, and 0.9. Figure 16 presents the budget-accuracy trade-off plots. We adopt a conservative setting of $\beta = 0.1$, as it consistently achieves strong performance. However, we note that even the $90\%$ quantile ($\beta = 0.9$) often yields reasonably high accuracy with significant budget savings, highlighting the flexibility of the approach.

Again, as also shown in Figures 12, 13, and 14, accuracy tends to increase monotonically with higher inference budget, thus a "sweet spot" doesn't exist. Exploring a principled metric to assess the cost-performance ratio within the inference-time scaling framework remains an interesting direction for future work.

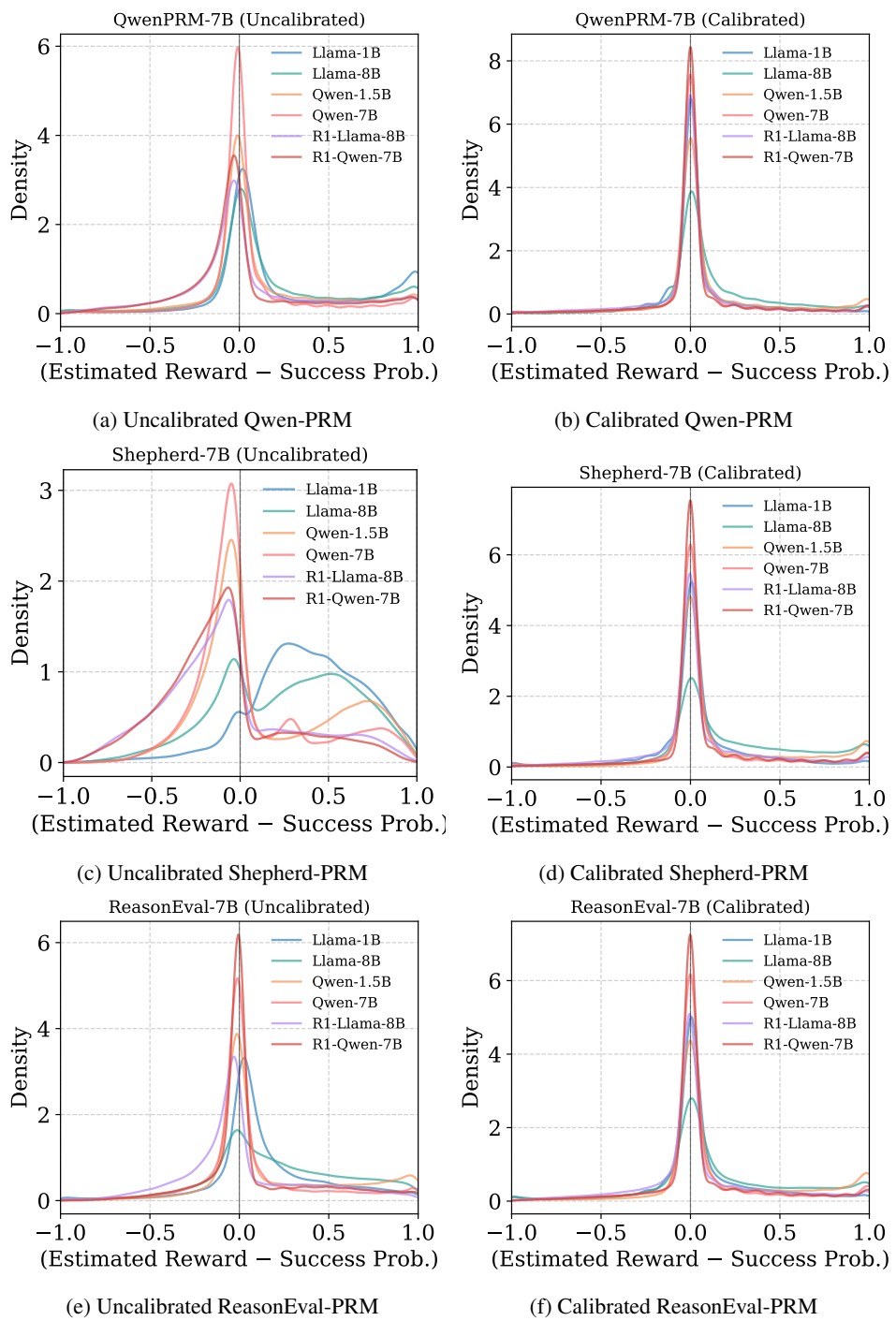

Figure 7: Histogram of signed deviation (i.e., estimation error) for Qwen, Shepherd, and ReasonEval PRMs evaluated on the MATH500 (in-distribution) datasets. Positive error indicates overestimation. As shown, **uncalibrated PRMs—the left column—tend to be optimistic (e.g., exhibiting higher density on the right side compared to the left and/or forming a noticeable peak around 1.0).** This becomes particularly pronounced for weaker models. In contrast, our calibration approach—the right column—transforms the error histograms into unbiased zero-mean distributions, consistently across various models.

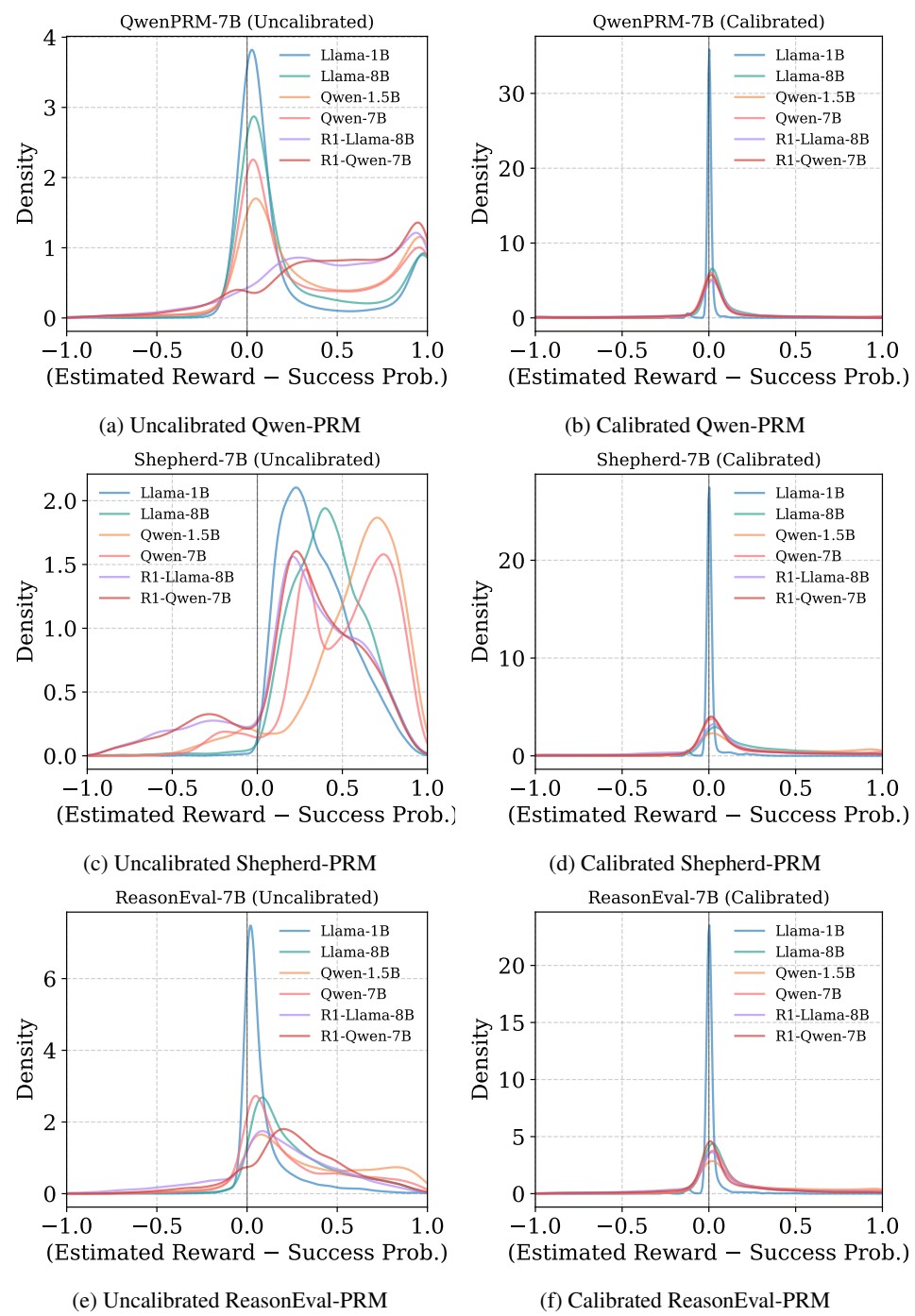

(a) Uncalibrated Qwen-PRM       (b) Calibrated Qwen-PRM

(c) Uncalibrated Shepherd-PRM       (d) Calibrated Shepherd-PRM

(e) Uncalibrated ReasonEval-PRM       (f) Calibrated ReasonEval-PRM

Figure 8: Histogram of signed deviation (i.e., estimation error) for Qwen, Shepherd, and ReasonEval PRMs evaluated on the `AIME24-25` (out-of-distribution) datasets. Positive error indicates overestimation. As shown, uncalibrated PRMs—the left column—tend to be optimistic (e.g., exhibiting higher density on the right side compared to the left and/or forming a noticeable peak around 1.0). **This becomes particularly pronounced for more challenging, out-of-distribution problems.** In contrast, our calibration approach—the right column—transforms the error histograms into unbiased zero-mean distributions, consistently across various models and dataset distributions.

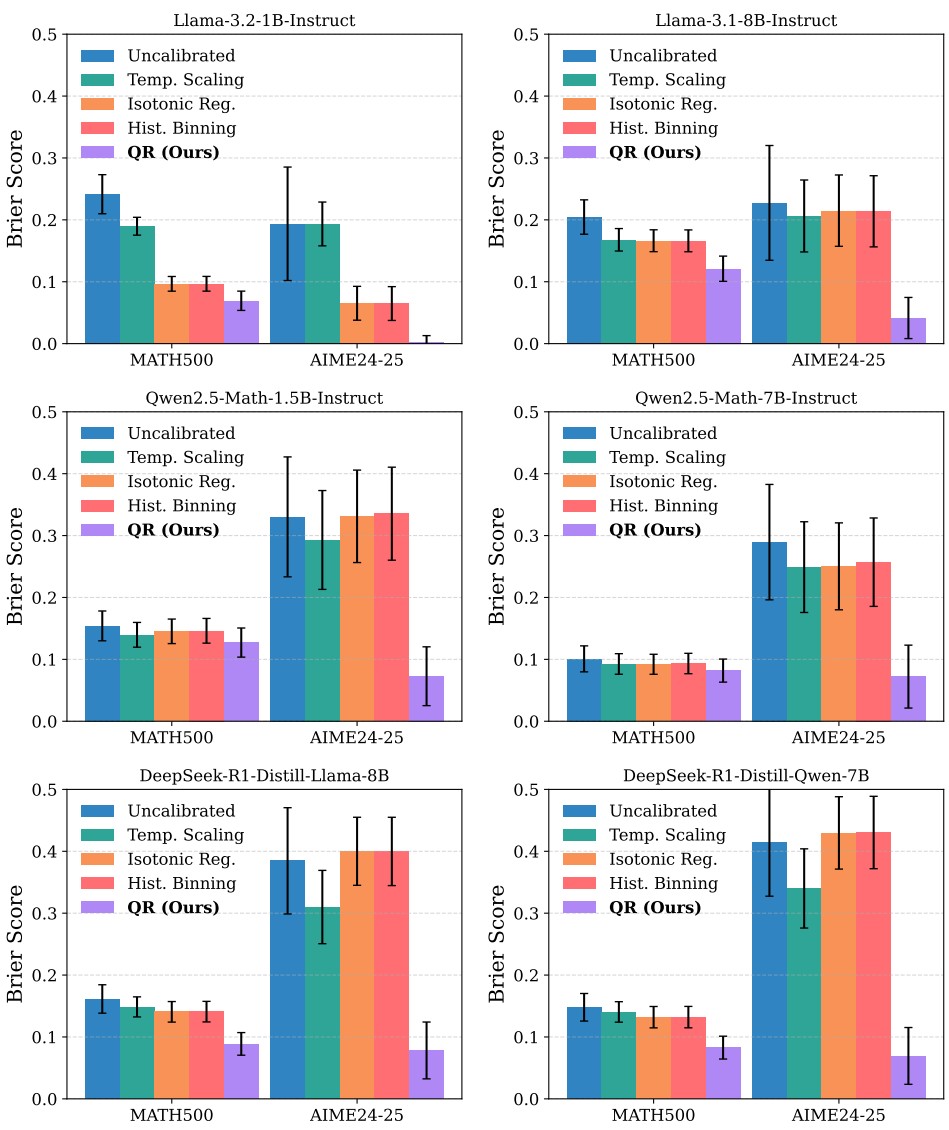

Figure 9: Comparison of our method with popular calibration techniques—temperature scaling, isotonic regression, and histogram binning—on the **Qwen** PRM across `MATH500` and `AIME24-25`.

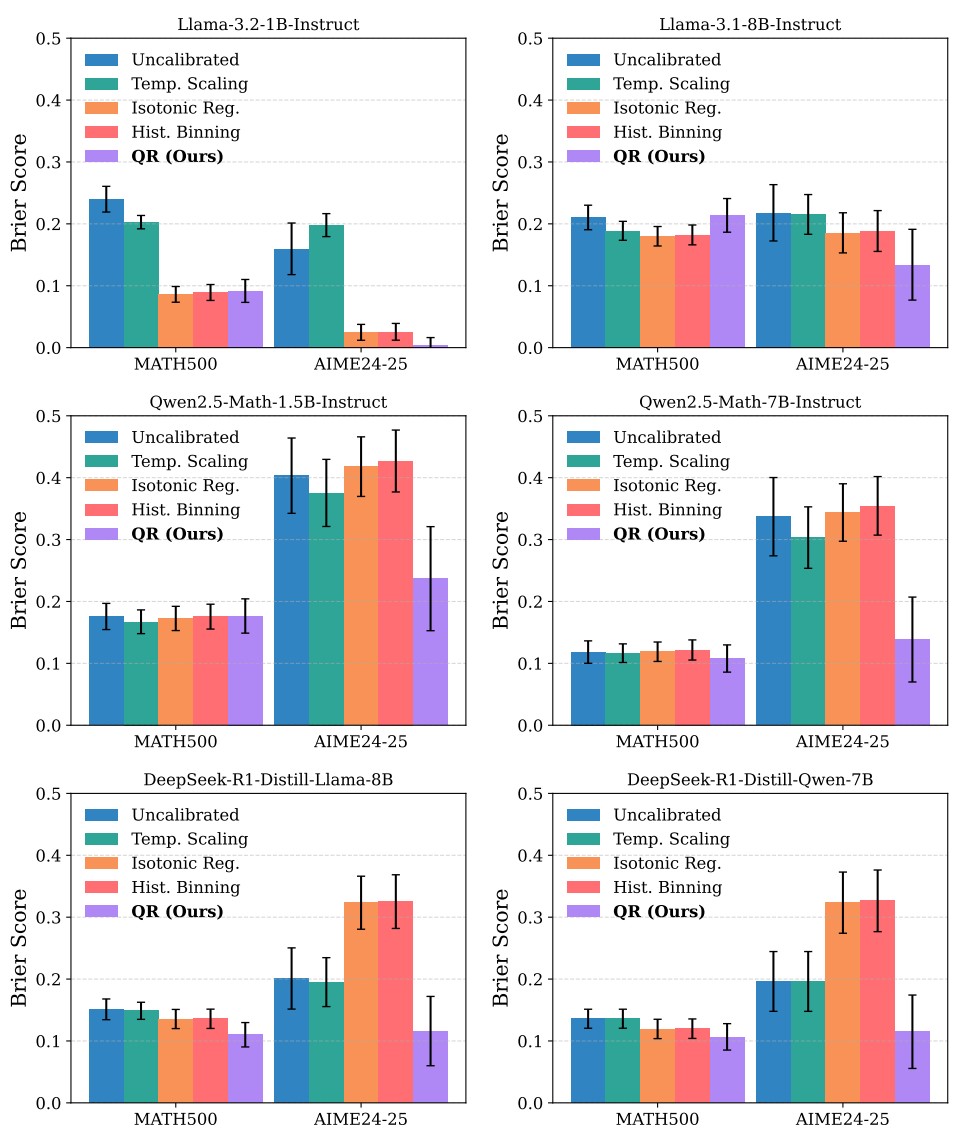

Figure 10: Comparison of our method with popular calibration techniques—temperature scaling, isotonic regression, and histogram binning—on the **Shepherd** PRM across `MATH500` and `AIME24-25`.

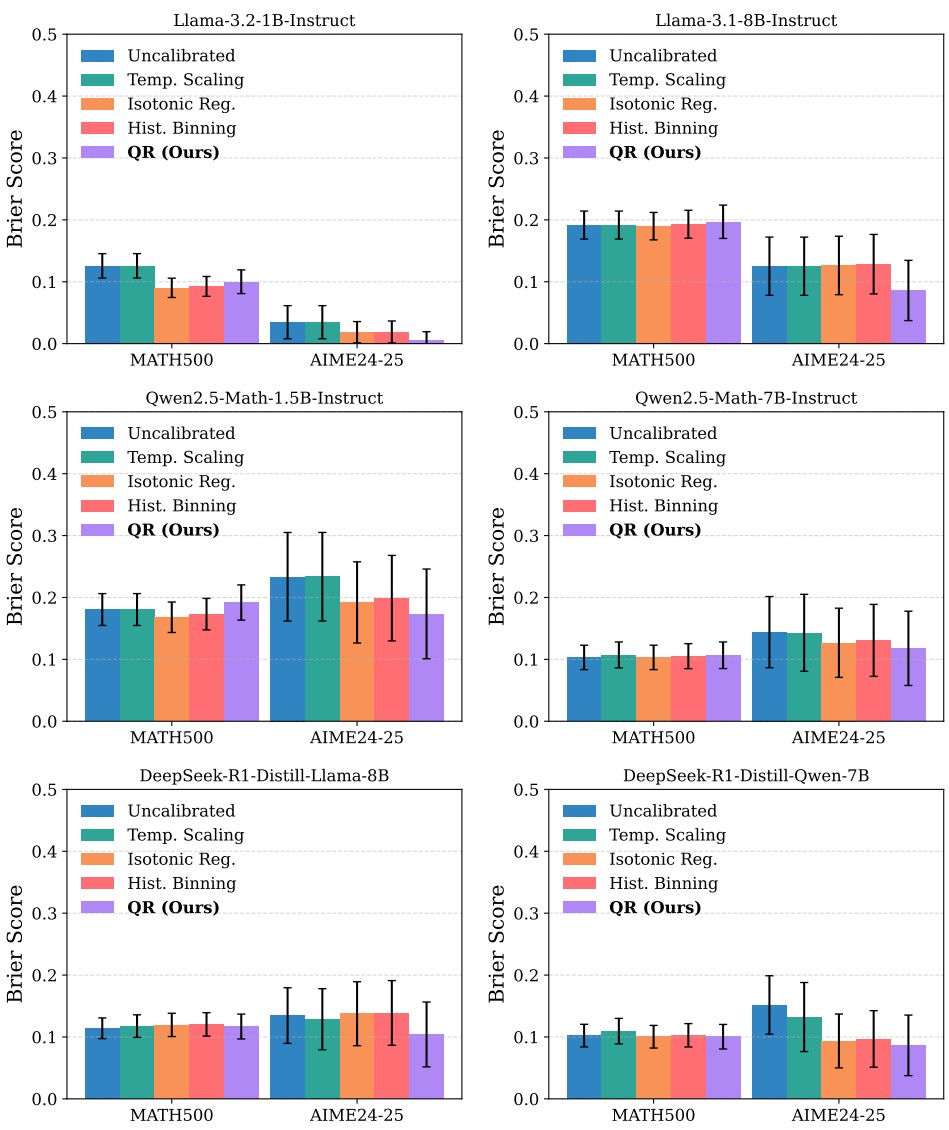

Figure 11: Comparison of our method with popular calibration techniques—temperature scaling, isotonic regression, and histogram binning—on the **ReasonEval** PRM across `MATH500` and `AIME24-25`.

Table 6: Evaluation of the best-of-$N$ method with a fixed-$N$ strategy (BoN), compared against the proposed instance-adaptive sampling strategy (BoN+IAS), with calibrated **Shepherd** PRM. We report both accuracy and normalized computational cost (budget). Relative improvements over Pass@1 accuracy are highlighted in light blue.

| Dataset | Model | Baselines | | w/ Uncal. PRM | | w/ Calib. PRM | |
|---|---|---|---|---|---|---|---|
| | | Pass@1 | BoN | BoN+IAS | Budget | BoN+IAS | Budget |
| MATH500 | Llama-3.2-1B | 0.2255 | 0.3800 | 0.3205 | 0.0852 | 0.3633 | 0.6379 |
| | Llama-3.1-8B | 0.4659 | 0.5620 | 0.5419 | 0.0852 | 0.5446 | 0.2481 |
| | Qwen-2.5-1.5B | 0.6970 | 0.7740 | 0.7518 | 0.0852 | 0.7414 | 0.1223 |
| | Qwen-2.5-7B | 0.7994 | 0.8460 | 0.8437 | 0.0852 | 0.8345 | 0.1048 |
| | R1-Llama-8B | 0.6734 | 0.7640 | 0.7493 | 0.0852 | 0.7767 | 0.3106 |
| | R1-Qwen-7B | 0.7556 | 0.8480 | 0.8137 | 0.0852 | 0.8206 | 0.2477 |
| AIME24-25 | Llama-3.2-1B | 0.0042 | 0.0000 | 0.0067 | 0.1206 | 0.0000 | 0.9956 |
| | Llama-3.1-8B | 0.0268 | 0.0333 | 0.0442 | 0.1206 | 0.0355 | 0.7320 |
| | Qwen-2.5-1.5B | 0.0932 | 0.1500 | 0.1310 | 0.1206 | 0.1397 | 0.6159 |
| | Qwen-2.5-7B | 0.0885 | 0.2000 | 0.1497 | 0.1206 | 0.1507 | 0.5740 |
| | R1-Llama-8B | 0.0784 | 0.1333 | 0.1505 | 0.1206 | 0.1760 | 0.8664 |
| | R1-Qwen-7B | 0.1411 | 0.3000 | 0.2263 | 0.1206 | 0.2755 | 0.8456 |

Table 7: Evaluation of the best-of-$N$ method with a fixed-$N$ strategy (BoN), compared against the proposed instance-adaptive sampling strategy (BoN+IAS), with calibrated **ReasonEval** PRM. We report both accuracy and normalized computational cost (budget). Relative improvements/degradations over Pass@1 accuracy are highlighted in light blue/red.

| Dataset | Model | Baselines | | w/ Uncal. PRM | | w/ Calib. PRM | |
|---|---|---|---|---|---|---|---|
| | | Pass@1 | BoN | BoN+IAS | Budget | BoN+IAS | Budget |
| MATH500 | Llama-3.2-1B | 0.2255 | 0.2900 | 0.2887 | 0.0773 | 0.3095 | 0.4672 |
| | Llama-3.1-8B | 0.4659 | 0.4780 | 0.5055 | 0.0773 | 0.5007 | 0.2570 |
| | Qwen-2.5-1.5B | 0.6970 | 0.7340 | 0.7400 | 0.0773 | 0.7286 | 0.1324 |
| | Qwen-2.5-7B | 0.7994 | 0.7500 | 0.8129 | 0.0773 | 0.7948 | 0.1523 |
| | R1-Llama-8B | 0.6734 | 0.5300 | 0.6335 | 0.0773 | 0.6142 | 0.2069 |
| | R1-Qwen-7B | 0.7556 | 0.6280 | 0.7194 | 0.0773 | 0.7016 | 0.1653 |
| AIME24-25 | Llama-3.2-1B | 0.0042 | 0.0000 | 0.0050 | 0.1521 | 0.0000 | 0.9852 |
| | Llama-3.1-8B | 0.0268 | 0.0333 | 0.0287 | 0.1521 | 0.0370 | 0.8773 |
| | Qwen-2.5-1.5B | 0.0932 | 0.0833 | 0.0955 | 0.1521 | 0.0762 | 0.7430 |
| | Qwen-2.5-7B | 0.0885 | 0.0500 | 0.1117 | 0.1521 | 0.0647 | 0.8471 |
| | R1-Llama-8B | 0.0784 | 0.0333 | 0.0388 | 0.1521 | 0.0238 | 0.8263 |
| | R1-Qwen-7B | 0.1411 | 0.0000 | 0.0807 | 0.1521 | 0.0468 | 0.8333 |

Table 8: Evaluation of the beam search method with a fixed-$N/M$, compared against the proposed adaptive sampling strategy (IASoK and IASoM) using calibrated **Shepherd** PRMs. We report both accuracy and computational cost (budget), measured by the average number of LLM generations per question normalized by that of a fixed-budget BS baseline. Relative improvements over the Pass@1 accuracy are highlighted in light blue.

| | | Baselines | | IAS w/ Calibrated PRM | | | |
| Dataset | Model | Pass@1 | BS | BS+IASoK | Budget | BS+IASoM | Budget |
| --- | --- | --- | --- | --- | --- | --- | --- |
| MATH100 | Llama-3.2-1B | 0.2255 | 0.4000 | 0.4000 | 0.9549 | 0.4300 | 1.0634 |
| | Llama-3.1-8B | 0.4659 | 0.5700 | 0.5400 | 0.5377 | 0.5600 | 0.5902 |
| | Qwen-2.5-1.5B | 0.6970 | 0.7900 | 0.7900 | 0.4806 | 0.7800 | 0.5014 |
| | Qwen-2.5-7B | 0.7994 | 0.7800 | 0.8900 | 0.5382 | 0.8900 | 0.5794 |
| | R1-Llama-8B | 0.6734 | 0.7900 | 0.8200 | 0.5264 | 0.8300 | 0.5672 |
| | R1-Qwen-7B | 0.7556 | 0.8600 | 0.8800 | 0.4063 | 0.9000 | 0.4983 |
| AIME24 | Llama-3.2-1B | 0.0078 | 0.0333 | 0.0333 | 1.0000 | 0.0333 | 1.0000 |
| | Llama-3.1-8B | 0.0479 | 0.1000 | 0.1000 | 0.8689 | 0.0667 | 0.9636 |
| | Qwen-2.5-1.5B | 0.0969 | 0.1000 | 0.1000 | 0.7565 | 0.1000 | 0.8523 |
| | Qwen-2.5-7B | 0.0979 | 0.1333 | 0.1333 | 0.7994 | 0.1333 | 0.9051 |
| | R1-Llama-8B | 0.0656 | 0.0000 | 0.1000 | 0.8958 | 0.1000 | 0.9065 |
| | R1-Qwen-7B | 0.1354 | 0.1000 | 0.3333 | 0.9917 | 0.3000 | 1.0713 |

Table 9: Evaluation of the beam search method with a fixed-$N/M$, compared against the proposed adaptive sampling strategy (IASoK and IASoM) using calibrated **ReasonEval** PRMs. We report both accuracy and computational cost (budget), measured by the average number of LLM generations per question normalized by that of a fixed-budget BS baseline. Relative improvements/degradations over Pass@1 accuracy are highlighted in light blue/red.

| | | Baselines | | IAS w/ Calibrated PRM | | | |
| Dataset | Model | Pass@1 | BS | BS+IASoK | Budget | BS+IASoM | Budget |
| --- | --- | --- | --- | --- | --- | --- | --- |
| MATH100 | Llama-3.2-1B | 0.2255 | 0.3400 | 0.4000 | 0.5030 | 0.3700 | 0.6730 |
| | Llama-3.1-8B | 0.4659 | 0.5700 | 0.5700 | 0.3606 | 0.5700 | 0.4490 |
| | Qwen-2.5-1.5B | 0.6970 | 0.7800 | 0.7700 | 0.4171 | 0.7900 | 0.4281 |
| | Qwen-2.5-7B | 0.7994 | 0.8800 | 0.8700 | 0.4664 | 0.8700 | 0.4840 |
| | R1-Llama-8B | 0.6734 | 0.3500 | 0.7900 | 0.3379 | 0.8300 | 0.4088 |
| | R1-Qwen-7B | 0.7556 | 0.4800 | 0.8900 | 0.3173 | 0.8900 | 0.3628 |
| AIME24 | Llama-3.2-1B | 0.0078 | 0.0000 | 0.0000 | 0.9258 | 0.0000 | 0.9782 |
| | Llama-3.1-8B | 0.0479 | 0.0333 | 0.0000 | 0.4007 | 0.0333 | 0.7371 |
| | Qwen-2.5-1.5B | 0.0969 | 0.1333 | 0.1000 | 0.3134 | 0.0667 | 0.5084 |
| | Qwen-2.5-7B | 0.0979 | 0.1000 | 0.1000 | 0.3626 | 0.0667 | 0.5155 |
| | R1-Llama-8B | 0.0656 | 0.0667 | 0.1000 | 0.7205 | 0.2000 | 0.8866 |
| | R1-Qwen-7B | 0.1354 | 0.2000 | 0.2333 | 0.6203 | 0.3333 | 0.6861 |

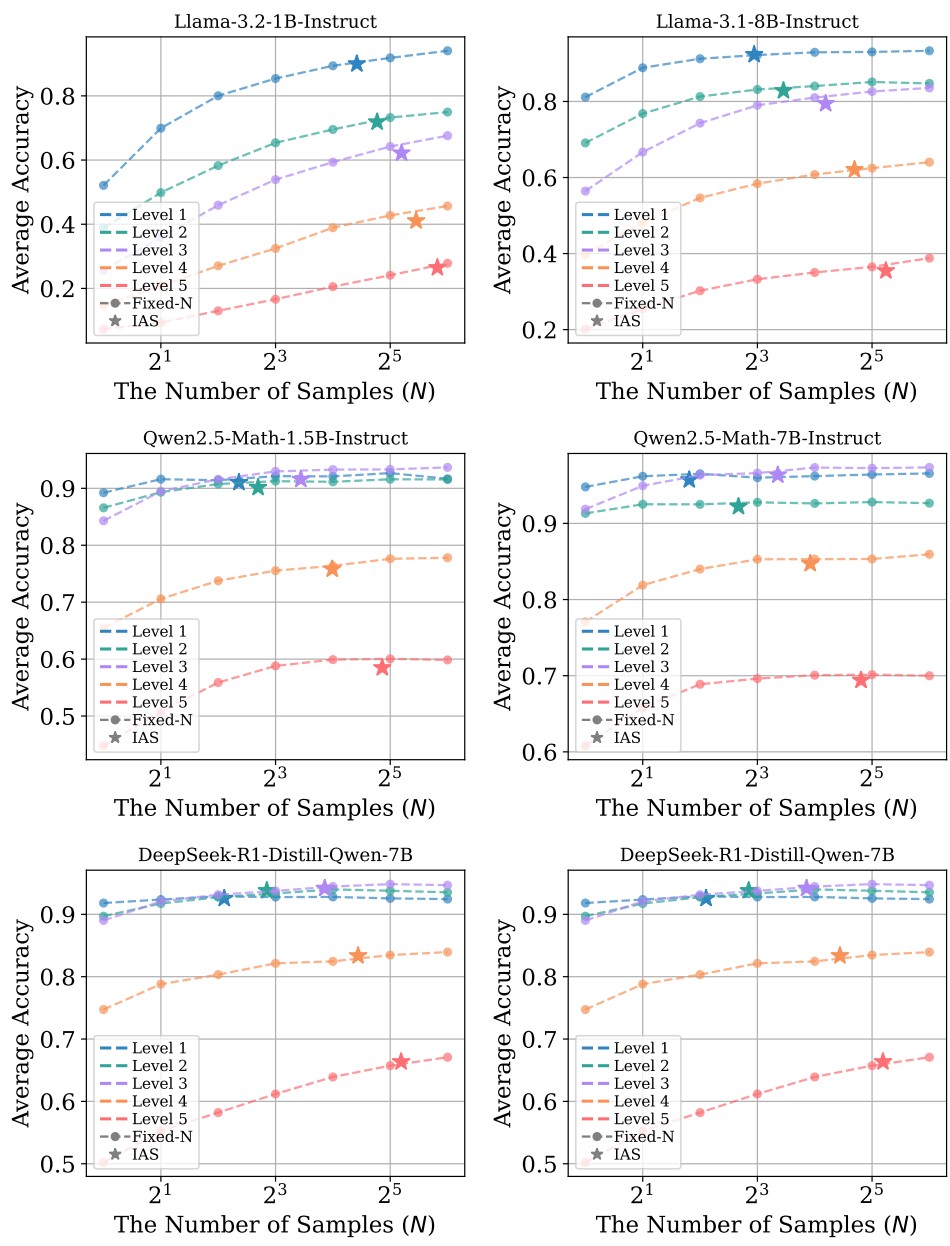

Figure 12: IAS scales with problem difficulty in `MATH500`, with **Qwen** PRM. Average accuracy over the test points with varying difficulty levels (1: easy, 5: hard) is illustrated. Regular fixed-$N$ and our IAS approaches are indicated by dashed lines and stars, respectively, and IAS allocates more samples to harder questions.

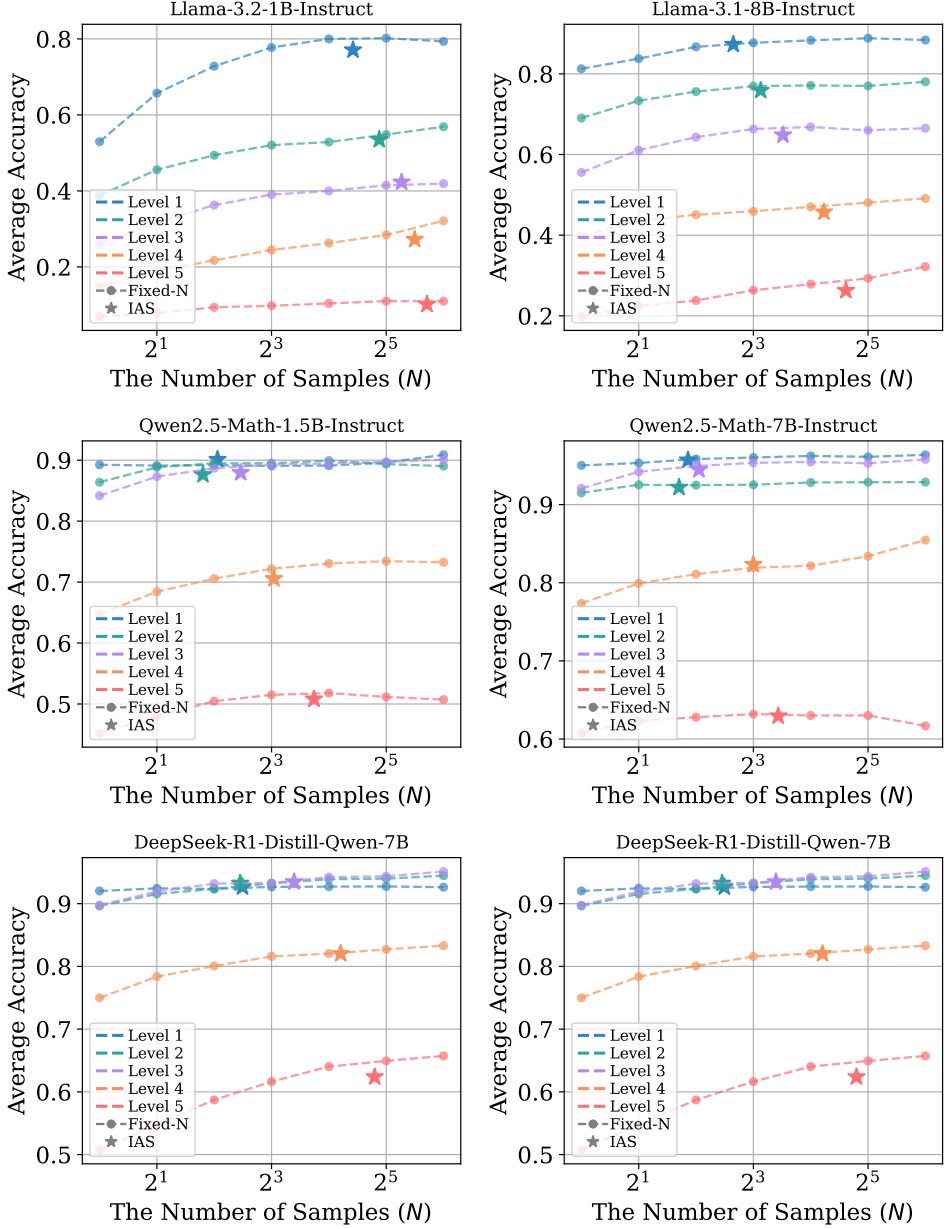

Figure 13: IAS scales with problem difficulty in `MATH500`, with **Shepherd** PRM. Average accuracy over the test points with varying difficulty levels (1: easy, 5: hard) is illustrated. Regular fixed-$N$ and our IAS approaches are indicated by dashed lines and stars, respectively, and IAS allocates more samples to harder questions.

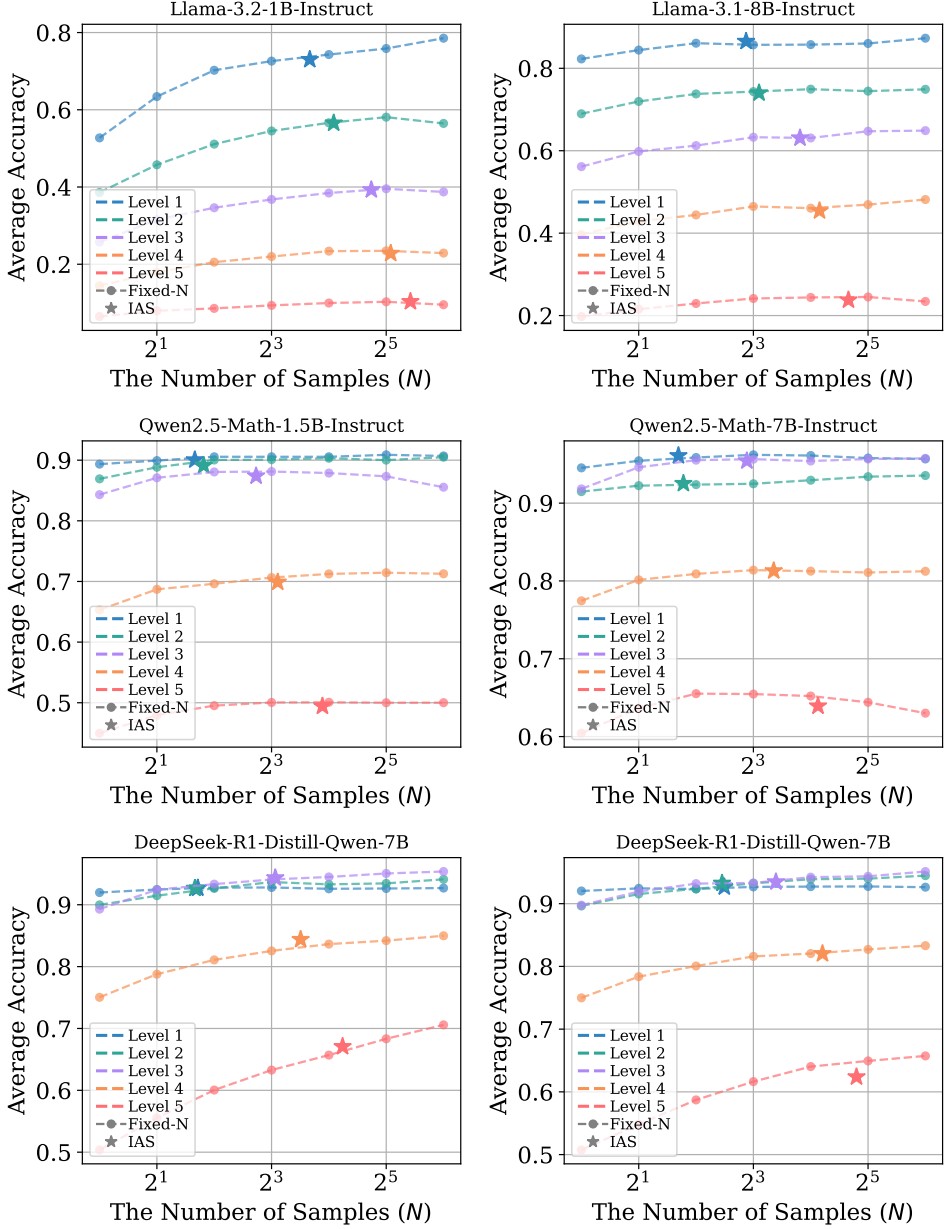

Figure 14: IAS scales with problem difficulty in `MATH500`, with **ReasonEval** PRM. Average accuracy over the test points with varying difficulty levels (1: easy, 5: hard) is illustrated. Regular fixed-$N$ and our IAS approaches are indicated by dashed lines and stars, respectively, and IAS allocates more samples to harder questions.

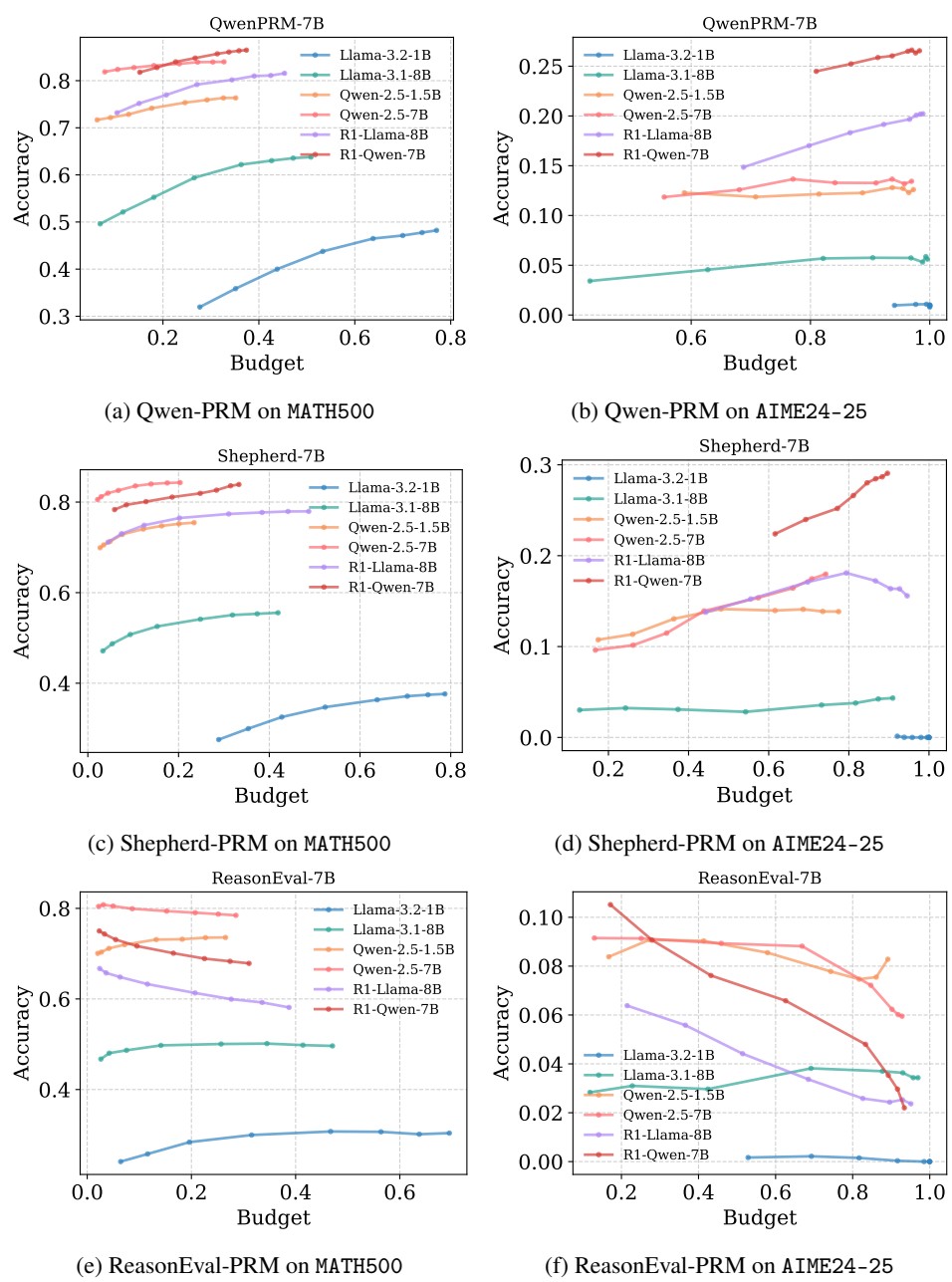

(a) Qwen-PRM on MATH500      (b) Qwen-PRM on AIME24-25

(c) Shepherd-PRM on MATH500      (d) Shepherd-PRM on AIME24-25

(e) ReasonEval-PRM on MATH500      (f) ReasonEval-PRM on AIME24-25

Figure 15: Budget vs. accuracy plots on MATH500 and AIME24-25 datasets with varying $C$.

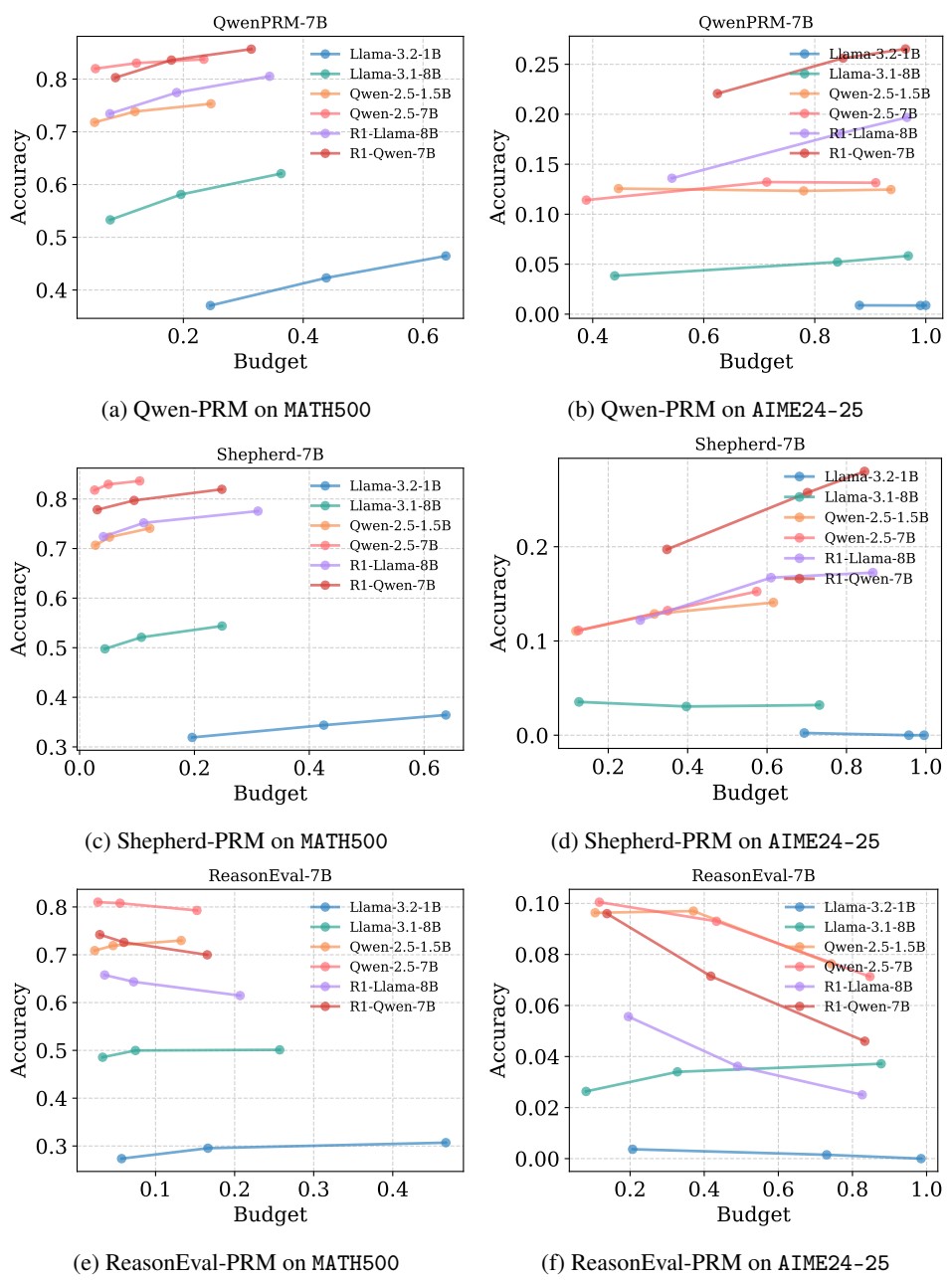

Figure 16: Budget vs. accuracy plots on `MATH500` and `AIME24-25` datasets with varying $\beta$.

