# OpenReview forum: "Know What You Don't Know: Uncertainty Calibration of Process Reward Models"
_NeurIPS.cc/2025/Conference — NeurIPS 2025 poster_

### Official Review · Reviewer_cDdM · 2025-06-20

**Clarity:** 3
**Significance:** 3
**Originality:** 4
**Rating:** 5
**Confidence:** 4

**Summary:**

The paper presents a novel framing of process reward model (PRM) predictions, when normalised, as approximations of the probability of generating a correct response. The authors demonstrate that current state-of-the-art PRMs produce miscalibrated forecast probabilities and argue why proper calibration is critical for efficient reasoning and for assessing problem difficulty on the fly. To address this, they introduce an adaptive compute-budget allocation method that leverages calibrated PRMs, along with theoretical justifications, to dynamically adjust evaluation time based on estimated difficulty. Experiments on two reasoning benchmarks across several models show promising gains in both calibration and compute efficiency.

**Questions:**

1. **Correctness measurement:** How did you determine whether a generated response was correct? I could not find a detailed description of the correctness metric in the paper.
2. **Sampling settings:** What sampling parameters (e.g., temperature, top-k or top-p) did you use when generating responses with multinomial sampling? More evaluation details would strengthen reproducibility.

**Ethical Concerns:**

["NO or VERY MINOR ethics concerns only"]

**Final Justification:**

Initially I was positive about the paper. In the rebuttal the authors have made significant attempts at clarifying and new experiments which has in my mind solidified the significance and impact of their paper. I am raising my score to 5 to justify this.

**Limitations:**

Yes

**Quality:**

3

**Strengths And Weaknesses:**

**Strengths**

1. **Clear motivation and presentation.** The paper is well motivated, and the core findings are presented in a logically structured, easy-to-follow manner.
2. **Novel insight on PRM normalisation.** Demonstrates that normalised PRM scores can be interpreted as forecast probabilities of success and that these are poorly calibrated out of the box for current SoTA PRMs.
3. **Evaluation of common recalibration techniques.** Shows that standard recalibration methods fail to properly calibrate PRMs on reasoning tasks. (It might be worth adding an additional baseline, adaptive temperature scaling, for completeness - see below.)
4. **Effective adaptive compute allocation.** The proposed method not only improves calibration but also aided in helping with compute-budget savings on easier problems whilst allocating more resources to harder questions rather than having a fixed sampling budget for sampling reasoning trajectories. This is a neat step toward more efficient reasoning with PRMs.

**Weaknesses**

1. **Notation and presentation clarity.** Some of the mathematical notation is a little sloppy, for example, Definition 1’s conditional probability should be written as p(x_{1:t}\mid q). The exposition of the quantile regression formulation (especially the first paragraph) is messy and needs rewriting for readability.
2. **Calibration metric choice.** In Figure 3, the Brier score is used to compare recalibration methods, but it conflates calibration and discrimination. A more standard calibration metric (e.g., Expected Calibration Error or Adaptive Calibration Error) would be preferable.
3. **Missing baseline.** While the paper correctly shows that conventional methods rescale or shift probabilities improperly, it should also compare against the adaptive temperature-scaling approach of Xie et al. 2024, which addresses some of these calibration challenges.
4. **Error bars**: Lack of error bars in the main results is somewhat limiting of the significance of the papers results. I think some effort should be made to change this given the high variance of MC sampling trajectories within some of their methods.

Xie, Johnathan, et al. "Calibrating Language Models with Adaptive Temperature Scaling." Proceedings of the 2024 Conference on Empirical Methods in Natural Language Processing. 2024.

---

> ### Author Rebuttal · Authors · 2025-07-30
>
> We thank Reviewer cDdM for their thoughtful feedback, valuable suggestions, and overall positive assessment of our work. Below, we address each of the questions raised, detailing how we will incorporate changes into the revised manuscript.
>
> ---
> ## Weaknesses
>
> ### W1 – Notation & presentation clarity
> >**Reviewer:** “Definition 1 should read $p(x_{1:t}\mid q)$; quantile-regression paragraphs need rewriting for readability.”
>
> **Response:** The current Definition 1 relies on the notation presented on page 3, lines 112–114:
> > Given a query $q$, the LLM generates a multi-step reasoning trajectory $x = (x_1, x_2, \dots, x_T)$, where $x_i$ denotes the $i$-th reasoning step and $T$ represents the total length of the trajectory. We further defined $x_{1:t}$ as the prefix of the trajectory up to step $t$.
>
> We clarify that $t$ refers specifically to an intermediate reasoning step, rather than the total length $T$. Thus, the expression $p(x_{1:t}\mid q)$ is an inaccurate presentation. Our intended definition is $p(x_{1:T} \text{ yields a correct answer} \mid q, x_{1:t})$. We acknowledge that this was not clearly articulated in the original manuscript. Additionally, we recognize that the phrase “yields a correct answer” may lack clarity. As per your suggestion in Question 1, we will explicitly specify our method for evaluating correctness.
>
> To improve the clarity and completeness of our manuscript, we will implement the following revisions:
>
> * **Make Section 4.1 Self-Contained**: We will merge the relevant content from the supplementary material (e.g., Appendix C.1) into Section 4.1 of the main text.
>
> * **Introduce a Dedicated Section for Data Collection**: We acknowledge that the procedure for collecting the calibration dataset is a key contribution. Therefore, we will create a new, independent section to describe this process in detail and provide pseudocode. The new section will formally outline our data collection method:
>
> 1.  For each question, we generate $N_{\text{val}}$ independent reasoning trajectories.
> 2.  For each reasoning trajectory and each corresponding prefix, we generate an additional $N_{\text{MC}}$ follow-up trajectories, conditioned on both the question and the prefix.
> 3.  We estimate the success probability by evaluating and counting the number of correct follow-up trajectories out of these $N_{\text{MC}}$ trajectories.
>
> ### W2 – Calibration metric choice
> > **Reviewer:** “Figure 3 uses Brier score; include ECE/AdaCE.”
>
> **Response:** We agree with your suggestion to include more conventional calibration metrics such as Expected Calibration Error (ECE) and Adaptive Calibration Error (AdaCE). Although initial space constraints limited the metrics presented in Figure 3, we have now conducted a comprehensive evaluation. Table R1 below provides detailed results for both uncalibrated and various calibration methods. Consistently, our method demonstrates effectiveness across all metrics tested. We empirically observe that AdaCE aligns better with the Brier score compared to ECE. This alignment is likely due to the non-uniform distribution of predictions between 0 and 1, which advocates the suitability of AdaCE over ECE as recommended in its original paper. We appreciate this suggestion, which has helped reinforce our findings, and we will explicitly include these expanded results in the final manuscript.
>
> ### W3 – Missing calibration method (Adaptive Temperature Scaling)
> > **Reviewer:** “Compare against ATS of Xie et al. 2024.”
>
> **Response:** We appreciate your recommendation to include the Adaptive Temperature Scaling (ATS) method proposed by Xie et al. (2024) as an additional calibration method to consider.
>
> We do point out that as-is, ATS was not proposed for and does not apply to PRM calibration, due to differences between typical LLMs’ language token outputs and the regression-style output of PRMs. That said, we were able to modify ATS to work for PRM calibration. Specifically, we adapted the ATS algorithm to train a transformer head that dynamically selects temperature parameters based on PRM-generated hidden states of the question and prefix. Note that this change makes it much closer to [1] than to the ATS paper. Below, we conduct experiments comparing our proposed method with this adaptation of ATS, and the results are detailed in Table R1. Key findings from these comparisons include:
> * The modified ATS method outperforms simpler baselines, reinforcing the importance of comprehensive, question- or prefix-aware calibration strategies (like our method).
> * For in-distribution datasets (e.g., MATH500), modified ATS achieves performance comparable to our LoRA fine-tuning approach. However, overall, our method consistently demonstrates greater effectiveness than modified ATS.
> * Crucially, our LoRA fine-tuning method significantly surpasses modified ATS in generalizing to challenging out-of-distribution datasets (e.g., AIME24-25), highlighting the robustness and broader applicability of our proposed approach.
>
> Combined with the advantages our quantile-based approach has for adaptive inference scaling, our proposed approach remains the best calibration method we consider. We will incorporate this detailed discussion into the revised manuscript and thank you again for this insightful suggestion.
>
> ### W4 – Error bars
> > **Reviewer:** “Add error bars to the main results.”
>
> **Response:** Thank you for highlighting the importance of including error bars. While error bars were provided in Figures 8, 9, and 10 in the supplementary materials, we will update figures in the main text (e.g., Figure 3) to clearly reflect the 95% confidence intervals.
>
> ---
> ## Questions
>
> ### Q1 – Correctness measurement
> > **Reviewer:** “How did you determine whether a generated response was correct?”
>
> **Response:** Thank you for requesting clarification. We utilized established validation procedures from prior works [2, 3]. In short, the generated answer within the `\boxed{}` is parsed and labeled correct if it matches the ground truth answer. Due to the nature of the math questions, there is little room for ambiguity. These details, along with relevant citations, will be explicitly included in the revised manuscript.
>
> ### Q2 – Sampling settings
> > **Reviewer:** “What sampling parameters (e.g., temperature, top-k or top-p) did you use?”
>
> **Response:** We appreciate your question regarding sampling parameters and reproducibility. Our experiments employed the widely used vLLM inference-acceleration framework [4] using their default sampling parameters (`top_p = 1.0`, `top_k = –1`; considering all tokens). The sampling temperature was set to 0.7, aligning with the standard practice recommended by prior works (e.g., Qwen, DeepSeek, various Llama variants, and scaling-law studies [3, 5]).
>
> We will provide a comprehensive section detailing these inference parameters, evaluation setups, and other relevant hyperparameters to enhance reproducibility. Additionally, we intend to publicly release our codebase alongside the final publication.
>
> ---
>
> Thank you again for the constructive review, your suggestions will certainly strengthen the paper. We hope these revisions and clarifications address all your concerns.
>
> ---
> ## References
> [1] Shen, Maohao, et al. "Thermometer: Towards universal calibration for large language models.", 2024.
> [2] Yang, An, et al. "Qwen2. 5-math technical report: Toward mathematical expert model via self-improvement.", 2024.
> [3] Beeching, Edward, and Tunstall, Lewis, and Rush, Sasha, "Scaling test-time compute with open models.", 2024.
> [4] Kwon, Woosuk, et al. "Efficient memory management for large language model serving with pagedattention.", 2023.
> [5] Puri, Isha, et al. "A probabilistic inference approach to inference-time scaling of llms using particle-based monte carlo methods.", 2025.
>
> ---
> ## Table
> **Table R1.** Calibration error for various methods and models on MATH500 (in-distribution) and AIME24-25 (out-of-distribution). Values are presented as **AdaCE**/ECE/Brier scores, where lower is better. The best result for each metric within a row is **bolded**.
>
> | Dataset | Model | Uncal. | TS | IR | HB | ATS | Ours |
> | :--- | :--- | :--- | :--- | :--- | :--- | :--- | :--- |
> | **MATH500** | L3.2-1B | .283/.279/.241 | .265/.268/.190 | .212/**.092**/.097 | .213/.093/.097 | .181/.151/.099 | **.081**/.094/**.069** |
> | | L3.1-8B | .263/.237/.205 | .244/.244/.168 | .321/.192/.166 | .323/.193/.166 | .237/.186/.130 | **.167**/**.152**/**.121** |
> | | Q2.5-1.5B | .218/.155/.154 | .205/.208/.140 | .271/**.117**/.145 | .274/.117/.146 | .205/.144/**.123** | **.155**/.141/.127 |
> | | Q2.5-7B | .146/.098/.101 | .135/.136/.093 | .201/**.018**/.092 | .205/.021/.094 | .135/.053/**.076** | **.100**/.086/.082 |
> | | R1-L-8B | .251/.131/.161 | .223/.224/.149 | .297/.022/.141 | .298/**.017**/.141 | .196/.063/.107 | **.113**/.091/**.089** |
> | | R1-Q-7B | .241/.110/.148 | .202/.206/.140 | .289/.019/.132 | .289/**.019**/.132 | .178/.069/.103 | **.107**/.086/**.083** |
> | **AIME24-25** | L3.2-1B | .236/.236/.194 | .210/.214/.193 | .177/.177/.065 | .177/.177/.065 | .125/.129/.065 | **.011**/**.004**/**.003** |
> | | L3.1-8B | .284/.284/.227 | .248/.258/.206 | .392/.392/.215 | .396/.396/.215 | .281/.281/.125 | **.086**/**.078**/**.041** |
> | | Q2.5-1.5B | .401/.401/.330 | .393/.410/.293 | .477/.477/.331 | .486/.486/.337 | .367/.346/.184 | **.105**/**.087**/**.073** |
> | | Q2.5-7B | .355/.355/.289 | .354/.368/.249 | .390/.390/.250 | .404/.404/.261 | .271/.259/.146 | **.098**/**.098**/**.072** |
> | | R1-L-8B | .526/.476/.385 | .528/.531/.310 | .590/.541/.400 | .588/.539/.399 | .438/.365/.220 | **.128**/**.076**/**.078** |
> | | R1-Q-7B | .558/.508/.414 | .570/.571/.340 | .613/.561/.430 | .613/.562/.430 | .440/.379/.230 | **.090**/**.069**/**.069** |
>
> **Acronyms**: Uncal. (Uncalibrated), TS (Temperature Scaling), IR (Isotonic Regression), HB (Histogram Binning), ATS (Adaptive Temperature Scaling), L (Llama), Q (Qwen).

---

> > ### Comment · Reviewer_cDdM · 2025-08-04
> >
> > Thank you for your detailed response, and I appreciate all the effort that you've made to address and investigate the comments that I had. I'm pretty satisfied with all the comments and will raise my score appropriately.
> >
> > One followup question that would help clarify things for me:
> >  - How do you deal with responses where the model did not produce an answer or your extraction method (looking for boxed) didn't find an answer? Playing around with reasoning models, this has happened for me a reasonable amount especially if one limits the number of new tokens. Were these excluded from evaluation? Might be good to talk about this a little bit in the paper for reproducibility.

---

> > > ### Author Response · Authors · 2025-08-05
> > >
> > > Thanks for raising this important question!
> > >
> > > First of all, we set the maximum token length to 4,096 to allow for long generations. We also instruct the model to answer within the box by prompting:
> > >
> > >
> > > ```
> > > Regardless of the approach, always conclude with: Therefore, the final answer is: $\\boxed{answer}$.
> > > ```
> > >
> > > During the **LLM evaluation**, we followed the standard evaluation protocol [1] and kept all generations, including those that failed to produce complete answers. The correctness of each response was determined by an exact match between the model's boxed output and the ground-truth answer—that is, if no boxed answer was provided, the response was marked as incorrect. We applied this evaluation criterion consistently across all models, datasets, and baselines. Empirically, we observe that extending the token limit does not significantly impact final performance—the sequences that fail to produce a boxed answer due to token constraints often still result in incorrect answers, even when the context window is extended. This is because many exceptionally long responses tend to be incorrect, as observed by many other prior works [2, 3, 4].
> > >
> > > However, as you noted, some generations may not produce a boxed answer. To ensure the reliability of our **calibration**, we excluded these unboxed generations when estimating the success probability via Monte Carlo simulation. We did this because an unboxed answer doesn't necessarily mean the LLM or its reasoning failed, as the token limit may have cut off a correct response. Note that these unboxed cases occurred in about 5-10% among all instances, so their impact on the calibration results is not significant.
> > >
> > > We agree that this is a critical point for reproducibility, and we'll be sure to include these details in our final paper revision.
> > >
> > > [1]  Beeching, Edward, and Tunstall, Lewis, and Rush, Sasha, "Scaling test-time compute with open models." (2024).
> > >
> > > [2] Cuadron, Alejandro, et al. "The danger of overthinking: Examining the reasoning-action dilemma in agentic tasks." arXiv preprint arXiv:2502.08235 (2025).
> > >
> > > [3] Wu, Yuyang, et al. "When more is less: Understanding chain-of-thought length in llms." arXiv preprint arXiv:2502.07266 (2025).
> > >
> > > [4] Jin, Zhensheng, et al. "ReCUT: Balancing Reasoning Length and Accuracy in LLMs via Stepwise Trails and Preference Optimization." arXiv preprint arXiv:2506.10822 (2025).

---

> ### Comment · Reviewer_cDdM · 2025-08-05
>
> Thank you for you comments. I would like to see the details that the authors have kindly presented that I think are very important for reproducibility and clarity included in the final version of the paper. I also feel reporting coverage, so %response that produce an answer would be good in response to our last back and forth. With this I'm happy to raise my score. Thanks to the authors for their prompt and thorough responses.

---

> > ### Author Response · Authors · 2025-08-05
> >
> > We deeply appreciate your effort, prompt response, and valuable feedback. As suggested, we will incorporate the experimental details (including the boxed response coverage) and results discussed into our revised manuscript. Additionally, we will release our code to ensure reproducibility.

---

### Official Review · Reviewer_wTjL · 2025-06-29

**Clarity:** 2
**Significance:** 3
**Originality:** 2
**Rating:** 4
**Confidence:** 4

**Summary:**

This paper explores several ways of calibrating existing PRMs for a given LLM by tuning PRMs towards the success rates of the given LLM for any prefix solutions. Typically, given a prefix solution of a given LLM, they continue to sample n outputs with LLM, then calculate the success rate of sampled outputs, finally, they can run quantile regression over those data to calibrate an existing PRM. With the calibrated PRM, they further utilize those calibrated scores to dynamically picking size of N outputs in BoN setting, and beam size B in beam search setting.  Experiments on calibration tasks show significant improvements over un-calibrated baseline. With the guidance of those PRMs, they can achieve comparable results with lower budget in best of 64 settings.

**Questions:**

1. In Section 5.1, “To start with, we construct a calibration dataset by randomly sampling 500 questions from the MATH training set. We assess calibration performance using standard metrics:”. Does this mean the authors tune and report on training set? Evaluation results in Table 1 is on complete solution or still on prefixes?

2. It’s not clear to me how the authors select prefix solutions, just uniformly sample on each time T over all tokens or on the end of a step?

3. Why not train a PRM directly on the calibration data with success rate.

**Ethical Concerns:**

["NO or VERY MINOR ethics concerns only"]

**Final Justification:**

The authors' responses have addressed most of my concerns.

**Limitations:**

See weaknesses and questions.

**Quality:**

3

**Strengths And Weaknesses:**

Strengths

1. Model calibration is an important property in inference time, as it can guide the search of reasoning especially in shot-CoT reasoning tasks.

Weaknesses

1. Experiments are not well-designed, and the following questions must be addressed (see Section Questions).
2. The “budget” metric in Table 2 is a little misleading, as you also need to run PRMs to get the calibrated scores, which are not required in baseline system. Thus, speed and memory usage should be also reported in Table 2.
3. Beam search with LLMs in this paper is not well-defined.

---

> ### Author Rebuttal · Authors · 2025-07-30
>
> Thank you for your careful reading of our paper and thoughtful feedback!
>
> ---
> ## Weaknesses
>
> ### W1 – Experiment
>
> **Q1 – Details of Calibration Data Collection I**
> > **Reviewer:** “Does this mean the authors tune and report on training set? Evaluation results in Table 1 is on complete solution or still on prefixes?”
>
> **Response:** To clarify, the MATH dataset provides both training and testing splits. We tune PRMs only using samples from the **training** split and test PRMs on the **test** split of the MATH dataset. Likewise, AIME2024 and AIME2025 are not used during PRM fine-tuning. Thus, calibration (i.e. fine-tuning) and testing are performed on distinct datasets. We will revise the manuscript accordingly to clarify this point.
> The evaluation results presented in Table 1 reflect performance on the **test** calibration dataset--- which as noted above is distinct from the **train** calibration dataset---encompassing all prefixes: initial questions with no prefix ($t = 0$), intermediate reasoning steps ($1 \leq t < T$), and fully completed solutions ($t = T$). We detail our calibration data collection process in Appendix C.1, summarized briefly below:
> 1.  For each question, we generate $N_{\text{val}}=8$ independent reasoning trajectories using the LLM.
> 2.  For each reasoning trajectory and each corresponding prefix trajectory, we generate an additional $N_{\text{MC}}=8$ follow-up trajectories, conditioned on both the question and the prefix.
> 3.  We estimate the success probability by evaluating and counting the number of correct follow-up trajectories out of these $N_{\text{MC}}$ trajectories.
>
> Ultimately, for each PRM-LLM pair, we generate 500 questions, each with 8 trajectories containing between 10 and 20 steps, resulting in approximately **40,000 to 80,000 prefixes** analyzed. We have further clarified these details in the revised manuscript.
>
> ---
> **Q2 – Details of Calibration Data Collection II**
> > **Reviewer:** “How the authors select prefix solutions … uniformly sample on each time T over all tokens or on the end of a step”
>
> **Response:** Due to space constraints, these details are provided in Appendix C.1 of our supplementary material. Specifically, prefixes are systematically collected at the end of each intermediate reasoning **step ($x_t$)**. We will integrate additional clarifications regarding the calibration dataset collection process into the revised manuscript.
>
> ---
> **Q3 – Details of Proposed Calibration Method**
> > **Reviewer:** “Why not train a PRM directly on the calibration data with success rate.”
>
> **Response:** We clarify that our proposed method **does involve directly training** (finetuning) the PRM on the calibration data using success rates. Specifically, we fine-tune off-the-shelf PRMs using the Low-Rank Adaptation (LoRA) technique [1], where the training objective is defined by the quantile loss.
> We adopt LoRA to fine-tune PRMs mainly due to its efficiency in computation, memory requirements, and inference time. LoRA works by introducing low-rank adapters into the weight matrices of PRMs, resulting in only a minimal increase, approximately **0.0025%**, in the total number of model parameters in our experiments.
>
>
>
>
>
>
> ---
> ### W2 – Computational Efficiency of the Proposed Method
>
> **Q1 - Budget Metric**
> > **Reviewer:** “The “budget” metric in Table 2 is a little misleading, as you also need to run PRMs to get the calibrated scores, which are not required in baseline system.”
>
> **Response:** We clarify that PRM calls are present in all baselines as well, and in fact, for both BoN and BS, our approach requires only 1 extra PRM call (the one on the initial question). Hence, the incremental cost of our extra PRM invocation is negligible. Furthermore, properly implemented, PRM evaluations are a very small part of the overall pipeline cost (baseline or ours), since each PRM evaluation requires generating only 1 token, compared to the thousands of tokens being generated by the main LLM. **More specifically, the token generated by PRM is about 1% of the LLM generation per reasoning step, and our IAS’s extra +1 score token adds less than 0.1%.**
>
> Specific details follow:
> 1.  **Baseline systems already pay for PRMs.**
>     * **Best-of-$N$ sampling:** The baseline also issues $N$ PRM calls—one for each candidate hypothesis. Our IAS variant issues $N_{\text{IAS}} + 1$ calls, where $N_{\text{IAS}} \le N$. The +1 evaluates the empty prefix (question-only) so that IAS can determine $N_{\text{IAS}}$.
>     * **Beam search:** The baseline issues $M \times K$ PRM calls per reasoning step. IAS needs $M_{\text{IAS}} \times K + 1$ or $M \times K_{\text{IAS}} + 1$—again, only one extra call.
>
> Even in the worst case, IAS makes $\approx 1.5\\%$ more PRM calls than the baseline.
>
> 2.  **Why PRM cost is negligible.**
>     Inference-time scaling targets the most expensive part of decoding: **LLM generation**, not the single-token logits of a PRM.
>     * **Single-token vs. trajectory length.** Each PRM call emits exactly **one** token; a reasoning trajectory for tasks like MATH typically emits **hundreds**.
>     * **Cost scales linearly with $N$.** Increasing the number of generated trajectories inflates the budget far more than an extra 1-token PRM call.
>
> For these reasons, Table 2 reports the metric most relevant to practitioners: **trajectories that the LLM actually *generates***. The cost of PRM calls, while present, is minimal in comparison.
>
> To provide a concrete example, here is a cost analysis using OpenAI's GPT pricing ($\textdollar$2/1M input tokens, $\textdollar$8/1M output tokens), a 100-token question, and 1,000-token reasoning trajectories. As the table below illustrates, when IAS, even in the worst case, generates the same number of trajectories as the baseline (64), the overhead is a negligible **$\textdollar$0.0002** (a 0.03% increase). However, when IAS intuits that only 32 trajectories (i.e., 50%) are sufficient, it slashes the total cost by nearly 50%.
>
> | Cost Element | Baseline (64 traj.) | IAS (worst: 64 traj.) | IAS (in practice: 32 traj.) |
> |:---|:---:|:---:|:---:|
> | LLM Generation (Input) | $0.0002 | $0.0002 | $0.0002 |
> | LLM Generation (Output) | $0.5120 | $0.5120 | $0.2560 |
> | **LLM Generation (Total)** | **$0.5122** | **$0.5122** | **$0.2562** |
> | --- | --- | --- | --- |
> | PRM Scoring (Input) | $0.1408 | $0.1410 | $0.0706 |
> | PRM Scoring (Output) | $0.000512 | $0.000520 | $0.000264 |
> | **PRM Scoring (Total)** | **$0.1413** | **$0.1415** | **$0.0709** |
> | --- | --- | --- | --- |
> | **Grand Total** | **$0.6535** | **$0.6537** | **$0.3271** |
> | **% of Baseline Cost** | **100%** | **~100.03%** | **~50.1%** |
>
> ---
> **Q2 - Speed and Memory Usage**
> > **Reviewer:** “Speed and memory usage?”
>
> **Response:** Great question! LoRA fine-tuning increases the model size by only **0.0025%** and adds ≈ **0.1%** compute overhead.
>
> In terms of memory, we fine-tune the PRM with LoRA adapters, which insert low-rank matrices into the 7B-parameter model and increase its size by only about **0.0025%**. On the speed side, LoRA’s extra multiply-adds scale with $O(2rd)$ instead of the original $O(d^2)$. Using $r=2$ and $d=3584$, the resulting compute overhead is roughly **0.1%**. IAS itself needs at most one additional PRM evaluation, but that evaluation produces a single score token, whereas each reasoning trajectory contains hundreds of generated tokens, so the incremental runtime cost is well below the already small 0.1% figure. We appreciate your comment and will add these quantitative clarifications to the manuscript to make the computational efficiency of our method clear.
>
> ---
> ### W3 – Details of Beam Search with LLMs
> > **Reviewer:** “Beam search with LLMs in this paper is not well-defined.”
>
> **Response:** Beam search is introduced in Section 2.2, and the method is presented in more detail in the paper [2] we cite in the original manuscript.
>
> ---
> ## References
> [1] Hu, Edward J., et al. "Lora: Low-rank adaptation of large language models.", 2022.
> [2] Snell, Charlie, et al. "Scaling llm test-time compute optimally can be more effective than scaling model parameters.", 2024.

---

> > ### Author Response · Authors · 2025-08-05
> >
> > Thank you again for your thoughtful feedback! We hope our response has addressed all your questions. If you have any additional comments that can help us further improve the paper, please let us know. If our response has addressed all your concerns, we would be grateful if you could re-evaluate our work.

---

> > ### Comment · Reviewer_wTjL · 2025-08-06
> >
> > Thanks for the clarification and more experiments. The responses addressed my main concerns. I'll update my scores.

---

### Official Review · Reviewer_dLMo · 2025-07-03

**Clarity:** 3
**Significance:** 2
**Originality:** 2
**Rating:** 4
**Confidence:** 4

**Summary:**

This paper addresses the critical issue of poor calibration in Process Reward Models (PRMs), which are essential components in inference-time scaling algorithms for LLMs. The authors demonstrate that state-of-the-art PRMs systematically overestimate success probabilities, particularly for challenging problems and weaker models. To remedy this, they propose a calibration method based on quantile regression and introduce an instance-adaptive scaling (IAS) framework that dynamically adjusts computational budgets based on calibrated uncertainty estimates.

**Questions:**

1. Regarding Line 214, BoN typically involves predicting complete sequences, but according to Definition 2, we need to estimate probability $p$ based on already generated content. Given that sequences differ from each other, is the estimated probability $p$ based solely on the question? How do you obtain the reward for the entire trajectory from the calibrated PRM—do you predict directly at the final step or average across all steps?
2.  We observe that on the more challenging AIME24-25 tasks, BS+IASoM shows remarkable improvements over BS (e.g., R1-Qwen-7B improving from 0.1333 to 0.3167). Since calibration shouldn't affect ranking order, what is the source of such significant improvements?
3. How is the training data for calibration obtained? Could you provide more details about the Monte Carlo rollout process and the quality of the generated success probability labels?
4. The final chosen parameters are C=0.99 and β=10% (90th percentile), which are far from the median and quite conservative. Does this lead to prohibitively high computational overhead?

**Ethical Concerns:**

["NO or VERY MINOR ethics concerns only"]

**Final Justification:**

The authors have satisfactorily addressed my concerns through comprehensive responses. The additional evaluation on 72B models confirms calibration issues persist across scales, strengthening the paper's generalizability. While the work remains limited to mathematical reasoning and the exact mechanism behind ranking improvements requires further ablation studies, these are minor issues that don't diminish the core contributions. The paper presents a well-motivated quantile regression approach for PRM calibration with clear practical benefits for inference-time scaling. I maintain my positive assessment.

**Limitations:**

Yes, in Page9.

**Paper Formatting Concerns:**

1. Line 197，ERM training
2. Figures 14 and 15 don't seem to show the changes in $C$ and $\beta$.

**Quality:**

3

**Strengths And Weaknesses:**

## Strengths
1. The paper effectively demonstrates that existing PRMs are poorly calibrated through comprehensive empirical analysis.
2. The use of quantile regression for PRM calibration is creative and well-motivated, addressing the need for uncertainty-aware predictions rather than just point estimates.
3.The proposed IAS framework has immediate practical applications for making LLM inference more efficient.

## Weaknesses
1. The experimental results are primarily based on PRMs with around 7B parameters, lacking analysis of larger-scale PRMs. It remains unclear whether larger models like Qwen/Qwen2.5-Math-PRM-72B also exhibit the overestimation phenomenon observed in smaller models.
2. The evaluation is confined to the mathematical reasoning domain. While this may be due to current PRM limitations, with the emergence of more generalizable PRMs such as Gen-Verse/ReasonFlux-PRM-7B, it's uncertain whether the observations and methods can extend to broader domains beyond mathematics.
3. The paper lacks comprehensive analysis of the training process, including training data composition and methodology. Only brief mention of LoRA fine-tuning is provided in the appendix (page 24), leaving important implementation details unclear.

---

> ### Author Rebuttal · Authors · 2025-07-30
>
> We thank Reviewer dLMo for their thoughtful feedback, valuable suggestions, and overall positive assessment of our work. Below, we address each of the questions raised, detailing how we have incorporated changes into the revised manuscript.
>
> -----
> ## Weaknesses
>
> ### W1 – Larger-scale PRMs
>
> > **Reviewer:** “lacking analysis of larger-scale PRMs.”
>
> **Response:** We appreciate the suggestion and have evaluated the 72B model on MATH and AIME. Results show larger models also suffer from overestimation, though slightly less than the 7B model. This is expected, as both were trained for ranking quality, not calibration. In contrast, our calibrated 7B model is significantly better calibrated than both uncalibrated 7B and 72B models, reinforcing our findings. We will add these results to the revision. We also note the 7B model is more widely used (64k downloads vs. 890 for 72B last month). While resource constraints prevent fully calibrating the 72B model, we hope our work encourages such studies at scale.
>
> **Table R0.** Calibration errors. Values are presented as Brier/Positive Brier scores, where lower is better. The best result for each metric within a row is **bolded**. L: Llama, Q: Qwen.
>
> |Dataset|Model|Uncalibrated 7B|Uncalibrated 72B|Calibrated 7B|
> |-|-|-|-|-|
> |**MATH500**|L-1B|.241/.223|.211/.078|**.069**/**.047**|
> ||L-8B|.205/.177|.177/.111|**.121**/**.099**|
> ||Q-1.5B|.154/.130|.155/**.106**|**.127**/.107|
> ||Q-7B|.101/.085|.118/.057|**.082**/**.053**|
> ||R1-L-8B|.161/.114|.193/.072|**.089**/**.055**|
> ||R1-Q-7B|.148/.106|.176/.075|**.083**/**.058**|
> |**AIME24-25**|L-1B|.194/.192|.157/.063|**.003**/**.001**|
> ||L-8B|.227/.223|.183/.122|**.041**/**.035**|
> ||Q-1.5B|.330/.322|.256/.176|**.073**/**.053**|
> ||Q-7B|.289/.282|.205/.139|**.072**/**.066**|
> ||R1-L-8B|.385/.371|.237/.204|**.078**/**.030**|
> ||R1-Q-7B|.414/.402|.260/.215|**.069**/**.034**|
>
> ### W2 – Domains of application
>
> > **Reviewer:** “The evaluation is confined to the mathematical reasoning domain.”
>
> **Response:** We focused on mathematical reasoning as it is a robust testbed for LLM logical inference and the domain with the most developed PRMs (at the time of submission). While we agree that generalizing is the ultimate goal, resource constraints prevent exploring other domains here. In this regard, we parallel the large body of LLM reasoning literature, which to date has largely focused on math. Exploring the applicability of our approach to other domains, such as code generation, scientific QA, or commonsense reasoning, is a valuable and exciting avenue for future research, and we have updated the conclusion of our manuscript to explicitly acknowledge this limitation and highlight these potential future directions.
>
> ### W3 – Implementation Details
>
> > **Reviewer:** “The paper lacks an analysis of the training process, including training data composition and methodology.”
>
> **Response:** In the revision, we will add a more detailed description of the calibration dataset and fine-tuning process, which are crucial for reproducibility, expanding on the summary already in Appendix C.1 and C.3.
> Specifically, we will add:
>
>   * The specific prompt formats used for each PRM and LLM.
>   * Pseudo-code for the quantile regression head and the loss function.
>   * A detailed description of our correctness measurement method.
>   * Specifications and examples from the calibration dataset, including the total number of prefixes.
>
> To further support transparency and future research, we will also make the entire codebase and calibration dataset publicly available alongside the final publication.
>
> -----
> ## Questions
>
> ### Q1-1 – Clarification on Probability Estimation for BoN
>
> > **Reviewer:** “BoN typically involves predicting complete sequences … Is the estimated probability based solely on the question?”
>
> **Response:** Thank you for pointing this out. Your understanding is correct; in the case of BoN, the prefix generation is indeed the empty string. To make this more clear, we will revise Definition 1 by changing $x\_{1:t}$ to $x\_{0:t}$, while explicitly stating $x\_{0:0} := \\text{“”}$. Additionally, we will update Definition 2 to clearly indicate that the prefix can be the empty string.
>
> ### Q1-2 – Reward Calculation
>
> > **Reviewer:** “How do you obtain the reward for the entire trajectory from the calibrated PRM?”
>
> **Response:** The reward calculation follows the official implementation provided by each PRM developer, with the final score predicted at the last step in our experiments; as [1] and [2] also noted, using the *last* token often performs the best in the inference-time scaling regime. Specifically, for Qwen-PRM, the system prompt is:
>
> ```
> <|im_start|>system
> Please reason step by step, and put your final answer in \boxed{}.<|im_end|>
> ```
>
> The user prompt format is:
>
> ```
> <|im_start|>user
> ${Question}<|im_end|>
> <|im_start|>assistant
> ${Prefix}<extra_0><|im_end|><|endoftext|>
> ```
>
> The score is computed based on the token at the final step, which is `<extra_0>`. In the case of BoN, the `${Prefix}` is an empty string (`""`). We recognize that explicitly detailing the PRM reward calculation process improves clarity and readability. Accordingly, we will include this comprehensive explanation in the revised manuscript.
>
> ### Q2 – Improvements over Beam Search (BS)
>
> > **Reviewer:** “BS+IASoM shows remarkable improvements over BS. Since calibration shouldn't affect ranking order, what is the source of such significant improvements?”
>
> **Response:** Thank you for raising this insightful question. For standard baselines such as temperature scaling and isotonic regression, calibration indeed does not alter ranking order. However, our method involves fine-tuning the PRM on the calibration dataset, which can modify ranking order. While improved calibration does not inherently guarantee enhanced ranking performance, our empirical findings indicate that the proposed PRM calibration method positively impacts ranking accuracy, leading to the substantial improvements you observed.
>
> ### Q3 – Details of Calibration Data Collection
>
> > **Reviewer:** “How is the training data for calibration obtained? … quality of the generated success probability labels?”
>
> **Response:**
> * **Roll-out:** The details of our calibration data collection are provided in Appendix C.1. Briefly summarized:
>
> 1\. For each question, we generate $N\_{\\text{val}}=8$ independent reasoning trajectories using the LLM.
>
> 2\. For each reasoning trajectory and each corresponding prefix trajectory, we generate an additional $N\_{\\text{MC}}=8$ follow-up trajectories, conditioned on both the question and the prefix.
>
> 3\. We estimate the success probability by evaluating and counting the number of correct follow-up trajectories out of these $N\_{\\text{MC}}$ trajectories.
>
> We will update the main manuscript to enhance the readability.
>
> * **Label quality:** While the per-prefix MC estimator $\\hat p$ is unbiased, we acknowledge that a larger $N\_{\\text{MC}}$ would yield more statistically accurate estimates. Nevertheless, we collect **M = 40,000 to 80,000** prefixes per PRM and LLM pair. Aggregating over $M$ prefixes reduces the standard error of our calibration error (i.e., Brier score) down to:
> $$
> \approx \sqrt{\frac{2\sigma_\varepsilon^{4} + 4\sigma_\varepsilon^{2} \times \mathrm{MSE_{\text{true}}}}{M}} =  4.7\times10^{-4},
> \qquad
> \sigma_\varepsilon^{2} = \frac{p(1-p)}{N\_{\\text{MC}}} = 0.02 .
> $$
> where we use $M = 40,000$, $p=0.8$ and $\mathrm{MSE_{\text{true}}} = 0.1$ based on our empirical results.
>
> Hence, our reported average calibration performance metrics are statistically accurate enough, and the per-instance noise has a negligible impact on the global trends. That said, the instance-wise standard error with $N\_{\\text{MC}}=8$ does artificially increase the variance of the distribution we are modeling in our quantile regression objective. As a result, increasing $N\_{\\text{MC}}$ may improve results for our method. We were limited by computational resources.
>
> * **Resource trade-off:** Raising $N\_{\\text{MC}}$ reduces instance-level standard error as $1/\\sqrt{N\_{\\text{MC}}}$ while already require up to **640K** extra forward passes per PRM–LLM pair (we have a total of 18 pairs) even for the current setting of $N\_{\\text{MC}}=8$. This amount of inference takes about 150 hours on our V100 GPU. We will release the data/code so larger-scale replications can explore bigger $N\_{\\text{MC}}$.
>
> ### Q4 – Choice of IAS Parameters & Computational overhead
>
> > **Reviewer:** “The final chosen parameters are C=0.99 and β=10% (conservative) ... Does this lead to prohibitively high computational overhead?”
>
> **Response:** We would like to point out that we already show results on this regime in the paper. With these settings ($C=0.99$ and $\\beta=10\\%$), IAS reduces the sampling budget by up to approximately **65%** for the BoN strategy and **75%** for the BS strategy (Table 2 and 3) – in other words, providing large computational savings, not overhead. We don’t believe that this setting is as conservative as it sounds, because the union bound guarantee on end-to-end accuracy should be high, especially with the MATH benchmark. Put another way, our goal is to save unnecessary computation that does not meaningfully help performance, rather than aggressively trading off performance and compute.
>
> -----
> ## Format Concerns
>
> In Figures 14 and 15, more conservative settings (e.g., larger C) are indicated by points further to the right. The values tested are originally in Appendix E.4. We will clarify and improve both figures and captions to fully address your concerns.
>
> ERM refers to Empirical Risk Minimization; we will explicitly define this.
>
> -----
> ## References
> [1] Beeching, Edward, et al. "Scaling test-time compute with open models.", 2024.
> [2] Snell, Charlie, et al. "Scaling llm test-time compute optimally can be more effective than scaling model parameters.", 2024.

---

> > ### Comment · Reviewer_dLMo · 2025-08-04
> >
> > Thank you for your comprehensive and thoughtful rebuttal. I appreciate your clarification that "our method involves fine-tuning the PRM on the calibration dataset, which can modify ranking order." However, I remain somewhat puzzled about the underlying mechanism.
> > - Is this improvement primarily due to the additional training data exposure during calibration fine-tuning, or is there something inherent about the calibration objective (quantile regression) that enhances the model's ability to distinguish between solution quality?
> > - Have you conducted any ablation studies to isolate whether the performance gains come from: (a) simply seeing more training examples during calibration fine-tuning, versus (b) the specific calibration loss function encouraging better uncertainty estimation, which in turn improves ranking?

---

> > > ### Author Response · Authors · 2025-08-05
> > >
> > > We appreciate the reviewer's insightful question.
> > >
> > > We believe that the improvement is less likely to be from data exposure alone. Our calibration set of 40-80k trajectories is very small compared to the millions of steps used to train the original PRM (e.g., Qwen-PRM uses 4.5 million labels). Therefore, the performance gain cannot be explained simply by seeing more examples.
> > >
> > > Although the calibration procedure occasionally improved accuracy, particularly for R1 models, we note that this isn’t consistent across the other models and settings we consider. This is the expected outcome, since the primary goal of our calibration procedure is not to systematically improve ranking quality, but rather to produce reliable scores for effective instance-adaptive scaling (IAS). Hence, empirical evidence seems to contradict a potential hypothesis that calibrating PRM improves the ranking quality.
> > >
> > > That said, the observed performance difference for those beam search rows is potentially interesting given its size. To better explore this point, we will include the following ablation studies in our final revision:
> > >
> > > 1.  **Compare different fine-tuning strategies.** We will use the same appended datasets with a different loss function, such as cross-entropy loss (that does not explicitly impose the regression-style calibration). This will help us determine whether the gain is from the data or the objective function.
> > > 2.  **Compare calibrated PRM performance with and without IAS.** We will run the calibrated PRM with the full-N scaling (i.e., without the IAS). If its performance is similar to the IAS version, this will confirm that the IAS is working as intended and isn't introducing performance losses. Moreover, this experiment can confirm whether the ranking quality is improved or degraded after fine-tuning calibration.
> > > 3.  **Evaluate the impact of IAS on accuracy.** We will run inference-time scaling experiments using the base PRM for ranking and the calibrated PRM solely for selecting the IAS. This will help us prevent any false gains from fine-tuning from affecting the ranking process.

---

> > > > ### Comment · Reviewer_dLMo · 2025-08-06
> > > >
> > > > Thank you for your detailed and thoughtful response to my questions. The proposed ablation studies you outline—comparing different fine-tuning strategies, evaluating calibrated PRM performance with and without IAS, and isolating the impact of IAS on accuracy—will be valuable additions that should help clarify the source of performance improvements and strengthen the paper's contributions. I look forward to seeing these additional analyses in the final revision and maintain my positive assessment of this work, which makes important contributions to understanding and improving PRM calibration for efficient inference-time scaling.

---

> > > > > ### Author Response · Authors · 2025-08-06
> > > > >
> > > > > We truly appreciate your effort, prompt response, and valuable feedback. As suggested, we will incorporate the additional ablation results and discussions/results presented in the rebuttal into our revised manuscript.

---

### Note · Authors · 2025-08-12

Dear ACs and Reviewers,

Thank you for your service and thoughtful feedback throughout the review process. We’re pleased to see that our rebuttal successfully addressed the concerns raised and that the paper has been positively received by all reviewers. We believe the changes made have significantly improved the clarity and contribution of our work. We deeply appreciate your constructive input.

Sincerely,
The Authors

---

### Decision · Program_Chairs · 2025-09-17

**Decision:**

Accept (poster)

**Comment:**

This paper addresses the critical issue of poor calibration in PRMs, which systematically overestimate the probability of success when guiding inference-time scaling algorithms for LLMs. Its core contributions include: first, systematically demonstrating that even state-of-the-art PRMs (e.g., variants of Qwen and Llama PRMs) exhibit significant miscalibration, especially on challenging problems and weaker models; second, proposing a novel calibration method based on quantile regression. Fine-tuned via LoRA, this method adjusts raw PRM outputs into well-calibrated estimates of the true probability of correctness given a solution prefix, also providing confidence intervals. All three reviewers (dLMo, wTjL, cDdM) provided positive assessments while raising important constructive criticisms. The authors addressed nearly all concerns through detailed responses and commitments to revisions. By tackling an significant and underexplored problem in PRM calibration, the paper offers a highly original and practical contribution—combining an effective calibration technique with an efficient inference-time framework—supported by thorough experiments and convincing results. It thus fully meets the bar for acceptance at NeurIPS.